# InfoSEM: A Deep Generative Model with Informative Priors for Gene Regulatory Network Inference

**Tianyu Cui** [1]  **Song-Jun Xu** [1]  **Artem Moskalev** [1]  **Shuwei Li** [1]  **Tommaso Mansi** [1]  **Mangal Prakash** [1] [*]  **Rui Liao** [1] [*]

## Abstract

Inferring Gene Regulatory Networks (GRNs) from gene expression data is crucial for understanding biological processes. While supervised models are reported to achieve high performance for this task, they rely on costly ground truth (GT) labels and risk learning gene-specific biases—such as class imbalances of GT interactions—rather than true regulatory mechanisms. To address these issues, we introduce InfoSEM, an unsupervised generative model that leverages textual gene embeddings as informative priors, improving GRN inference without GT labels. InfoSEM can also integrate GT labels as an additional prior when available, avoiding biases and further enhancing performance. Additionally, we propose a biologically motivated benchmarking framework that better reflects real-world applications such as biomarker discovery and reveals learned biases of existing supervised methods. InfoSEM outperforms existing models by 38.5% across four datasets using textual embeddings prior and further boosts performance by 11.1% when integrating labeled data as priors.

## 1. Introduction

Gene Regulatory Networks (GRNs) govern cellular processes by capturing regulatory relationships essential for gene expression, differentiation, and identity (Karlebach & Shamir, 2008). Represented as directed graphs, they typically feature transcription factors (TFs) regulating target genes (TGs), with non-coding RNAs also playing key roles. GRNs have diverse applications, including mapping molecular interactions (Basso et al., 2005), identifying biomarkers (Dehmer et al., 2013), and advancing drug design (Ghosh & Basu, 2012).

[*]Shared last authors  [1]Johnson and Johnson Innovative Medicine. Correspondence to: Tianyu Cui <tcui8@its.jnj.com>.

*Proceedings of the $42^{nd}$ International Conference on Machine Learning*, Vancouver, Canada. PMLR 267, 2025. Copyright 2025 by the author(s).

Single-cell RNA sequencing (scRNA-seq) has revolutionized GRN inference by enabling high-resolution profiling of cell-specific regulatory interactions. However, scRNA-seq data are noisy, sparse, and high-dimensional (Pratapa et al., 2020; Wagner et al., 2016; Dai et al., 2024), requiring advanced computational approaches. Methods have evolved from co-expression frameworks (Chan et al., 2017; Kim, 2015) to cutting-edge machine learning (ML) and deep learning (DL) models (Yuan & Bar-Joseph, 2019; Wang et al., 2024; Anonymous, 2024; Shu et al., 2021), with further improvements leveraging complementary but hard to obatin perturbation experiments, RNA velocity, and chromatin accessibility inputs (Chevalley et al., 2022; Atanackovic et al., 2024; Yuan & Duren, 2024).

GRN inference methods can be broadly classified as supervised or unsupervised. Supervised models use experimentally derived ground truth (GT) labels, such as ChIP-seq data, where known TF-TG interactions guide learning. While they achieve high performance (AUPRC ∼0.85) (Chen & Liu, 2022; Wang et al., 2023), their reliance on costly GT labels limits applicability. Unsupervised methods which infer regulatory relationships directly from gene expression data without using any known interactions, present an attractive alternative but typically lag in performance compared to supervised models (Chen & Liu, 2022; Wang et al., 2023; Mao et al., 2023). Bridging this gap requires advancing unsupervised GRN inference methods.

To address this, we introduce InfoSEM, an unsupervised generative framework trained with variational Bayes that leverages textual gene embeddings as informative priors. By integrating prior biological knowledge, InfoSEM substantially improves GRN inference over existing models. Moreover, InfoSEM can incorporate GT labels as an additional prior when available, rather than using them for direct supervision, avoiding dataset (gene-specific) biases and improving performance further.

Beyond model development, reliably evaluating GRN inference methods is equally important. Existing supervised learning based GRN inference benchmarks typically assume all genes are represented in both the training and test sets, and only the regulatory links in the test set differ from those seen during training (*unseen interactions between*

*seen genes*, Section 4). While suitable for predicting interactions among well-characterized genes, this setup does not reflect many real-world applications such as biomarker discovery and rare cell-type studies (Lotfi Shahreza et al., 2018; Ahmed et al., 2020), which often involve genes whose interactions are completely absent from the training data (*interactions between unseen genes* in Section 4), thus requiring models to generalize beyond the set of genes encountered during training.

To bridge this gap, we propose a biologically motivated benchmarking framework evaluating *interactions between unseen genes*, better aligning with many real-world applications. This shift in evaluation perspective also inadvertently highlights that, in prior benchmarks, supervised models may have relied on gene-specific biases of the dataset, e.g., class imbalance in known interactions for each gene. This underscores the need for careful evaluation to distinguish between performance gains driven by genuine biological insights from scRNA-seq and those influenced by dataset biases.

In summary, our work makes the following contributions:

- We present InfoSEM, an unsupervised generative framework that leverages textual gene embeddings as priors and can seamlessly integrate GT labels as an additional prior (when available), avoiding dataset biases and improving GRN inference.

- We reveal limitations in existing supervised learning based GRN inference benchmarks, showing that supervised models may exploit dataset biases, such as class imbalance of each gene, rather than capturing true biological mechanisms from scRNA-seq.

- We propose a new evaluation framework focused on regulatory *interactions between unseen genes*, better aligning with real-world applications such as biomarker discovery.

- Finally, we demonstrate that our InfoSEM improves GRN inference by 38.5% over models without priors and achieves state-of-the-art performance, even surpassing supervised models. Integrating GT labels as an additional prior further boosts performance by 11.1% while mitigating dataset biases.

## 2. GRN inference problem

Let $G = (V, Y)$ represent a GRN, where $V = \{v_1, \ldots, v_P\}$ denotes the set of $P$ genes (nodes), including transcription factors (TFs) and target genes (TGs). The adjacency matrix $Y \in \{0, 1\}^{P \times P}$ encodes regulatory interactions, where $y_{ik} = 1$ indicates that TF $v_i$ regulates TG $v_k$. We assume access to scRNA-seq gene expression data and the goal is to infer the adjacency matrix $Y$, a setup commonly used in recent studies (Shu et al., 2021; Anonymous, 2024; Chen &

Liu, 2022; Chen & Zou, 2024; Haury et al., 2012; Moerman et al., 2019). The scRNA-seq gene expression data, comprising $P$ genes and $N$ cells, is represented as $X \in \mathbb{R}^{P \times N}$, where $x_{ij} \in \mathbb{R}$ denotes the expression of gene $v_i$ in cell $j$, and $\boldsymbol{x}_i$ is the $i$-th row of $X$, corresponding to the expression profile of $v_i$ across all cells.

### 2.1. Supervised GRN Inference

When partial information about the adjacency matrix $Y$ is available from sources like ChIP-seq or databases such as the Gene Transcription Regulation Database (Yevshin et al., 2019), GRN inference can be framed as a supervised learning (SL) task, where the model is trained on labeled data with known interactions in $Y$ serving as labels. The model learns to predict the probability of an interaction $y_{ik}$ being true based on observed data by minimizing a cross-entropy loss, i.e., assuming a Bernoulli likelihood. Specifically, SL methods aggregate representations of genes $i$ and $k$, denoted as $\boldsymbol{s}_i$ and $\boldsymbol{s}_k$, which could be a function of their expression profile $\boldsymbol{x}_i$ and $\boldsymbol{x}_k$, and predict the probability of interaction $y_{ik}$ using a function $f_{\text{sl}}$ on the aggregated representation:

$$p_{ik} = f_{\text{sl}}(\text{agg}(\boldsymbol{s}_i, \boldsymbol{s}_k)), y_{ik} \sim \text{Bernoulli}(p_{ik}). \quad (1)$$

Deep learning methods such as CNNC (Yuan & Bar-Joseph, 2019), DeepDRIM (Chen et al., 2021), and GENELink (Chen & Liu, 2022) leverage CNNs, GNNs, or transformers to aggregate pairwise gene expressions or their deep representations for interaction prediction (Wang et al., 2023; Mao et al., 2023; Xu et al., 2023). The SL framework can also flexibly incorporate pre-trained embeddings, such as BioBERT embeddings in scGREAT (Wang et al., 2024) or scBERT embeddings combined with GENELink representations in scTransNet (Kommu et al., 2024).

### 2.2. Unsupervised GRN Inference

Unsupervised learning (USL) methods infer gene relationships without labeled data, relying solely on gene expression data and more closely mirroring real-world scenarios (Pratapa et al., 2020) where labeled data is unavailable. Approaches include information theory-based methods (e.g., partial Pearson correlation (Kim, 2015), mutual information (Margolin et al., 2006), and PIDC (Chan et al., 2017)) and self-regression models, which predict a gene's expression based on the expressions of all other genes.

$$\boldsymbol{x}_i = f_i(\boldsymbol{x}_1, \ldots, \boldsymbol{x}_{i-1}, \boldsymbol{x}_{i+1}, \ldots, \boldsymbol{x}_P) + \boldsymbol{z}_i, \quad (2)$$

where $\boldsymbol{z}_i$ is Gaussian noise term. The feature importance of $\boldsymbol{x}_k$ in $f_i$ indicates the interaction effect from gene $k$ to $i$. TIGRESS (Haury et al., 2012) solves a linear regression for each $f_i$ in Eq.2,

$$X = A^T X + Z, \quad (3)$$

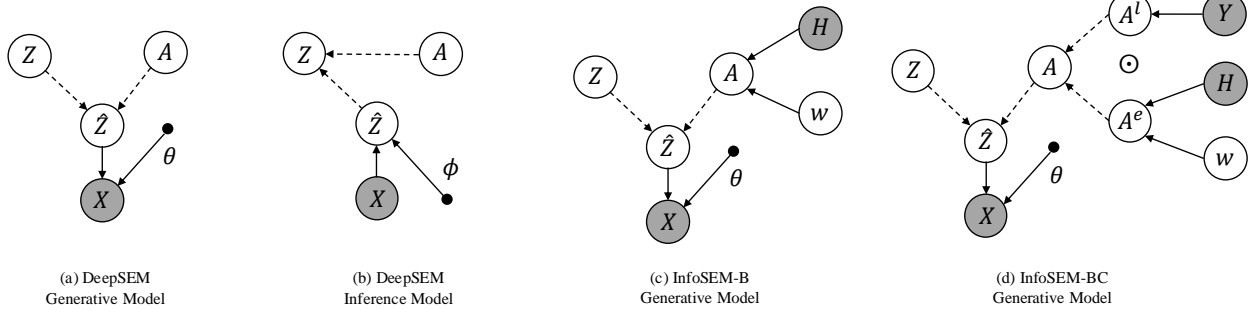

*Figure 1.* (a) Generative model of **DeepSEM**; (b) Inference model of **DeepSEM**; (c) Generative model of **InfoSEM-B**: InfoSEM with BioBERT gene-embedding priors $H$ on interaction effects; (d) Generative model of **InfoSEM-BC**: InfoSEM with both BioBERT gene-embedding priors on interaction effects $A^e$ and known interactions (e.g.-ChIP-seq) priors on logit of the probability of interactions $A^l$. Solid and dashed lines represent stochastic and deterministic relations respectively.

where all diagonal elements in the **weighted adjacency matrix** $A \in \mathbb{R}^{P \times P}$ are 0 and an $L_1$ regularization is applied on $A$. A post-hoc threshold, e.g., only keeping the top 1% interactions, can be further applied on weighted adjacency matrix $A$ to obtain adjacency matrix $Y$. GE-NIE3 (Huynh-Thu et al., 2010) and GRNBoost2 (Moerman et al., 2019) use random forests and gradientboosting to enhance model flexibility and feature importance representation. Equation 3 resembles the linear structural equation model (SEM) in Bayesian networks (Zheng et al., 2018), with an added acyclicity constraint on $A$. However, classical GRNBoost2 still outperforms even the latest Directed Acyclic Graphs (DAG) learning algorithms due to feedback effects (Chevalley et al., 2022). DeepSEM (Shu et al., 2021) extends Eq.3 by modeling gene expression in a latent space without the acyclicity constraint, using $\beta-$variational autoencoder (VAE) to denoise data, and outperforming traditional USL methods (Shu et al., 2021; Zhu & Slonim, 2024; Pratapa et al., 2020). However, it lacks support for incorporating external priors, which have shown to provide accuracy improvements for recent SL methods (Anonymous, 2024; Wang et al., 2024; Kommu et al., 2024).

## 3. InfoSEM: Informative priors for DeepSEM

Inspired by DeepSEM, we carefully design InfoSEM, a novel and principled generative model that incorporates multimodal informative priors: textual gene embeddings from pre-trained foundation models and known regulatory interactions (when available). InfoSEM is designed for scenarios both with and without labeled interactions, addressing limitations in the original DeepSEM framework. We begin by reviewing DeepSEM and then detail our approach to integrating these priors.

### 3.1. Introduction to DeepSEM

The linear SEM model in Eq. 3 can be viewed as a generative model: it generates the dataset $X$ with a GRN structure

specified by a weighted adjacency matrix $A$ by first generating random noise $Z$, and then solving $X = (\mathrm{I} - A^T)^{-1}Z$ (Yu et al., 2019). Similarly, DeepSEM defines a generative model, shown in Figure 1 (a), as follows:

$$A \sim p(A), \ Z \sim p(Z),$$
$$\hat{Z} = (\mathrm{I} - A^T)^{-1}Z, \ X \sim p_\theta(X|\hat{Z}), \qquad (4)$$

where the GRN structure, represented by $A$, is modeled on the latent space $\hat{Z}$ of $X$, rather than directly on $X$. Both $X$ and $\hat{Z}$ share the same GRN structure. We use $X \sim p_\theta(X|Z, A)$ to simplify the second line of the generative model in Eq.4. All distributions in Eq.4 are fully factorized:

$$p(A) = \prod_{i,k} \mathrm{Laplace}(a_{ik}; 0, \sigma_a),$$
$$p(Z) = \prod_{i,j} \mathcal{N}(z_{ij}; 0, \sigma_z^2),$$
$$p_\theta(X|\hat{Z}) = \prod_{i,j} \mathcal{N}(x_{ij}; f_\theta(\hat{z}_{ij}), g_\theta(\hat{z}_{ij})), \qquad (5)$$

where $a_{ik}$ represents the interaction effect from gene $i$ to gene $k$ and $z_{ij}$ is the corresponding latent variable of $x_{ij}$, i.e., the expression of gene $i$ in cell $j$. A Laplace prior is given to $A$ to encourage sparsity where the hyperparameter $\sigma_a$ controls the sparsity level. Similar to the vanilla VAE (Kingma, 2013), a zero-mean Gaussian prior with standard deviation $\sigma_z$ is applied to the latent variable $Z$ and both the mean and variance of the Gaussian likelihood of $X$ are given by the generation networks $f_\theta(\cdot)$ and $g_\theta(\cdot)$ parametrized by $\theta$. To learn DeepSEM, an inference network that approximates the posterior of $Z$, $Z \sim q_\phi(Z|X, A)$, is introduced as follows:

$$\hat{Z} \sim q_\phi(\hat{Z}|X), \ Z = (\mathrm{I} - A^T)\hat{Z}, \qquad (6)$$

where $q_\phi(\hat{Z}|X)$ is also fully factorized:

$$q_\phi(\hat{Z}|X) = \prod_{i,j} \mathcal{N}(\hat{z}_{ij}; f_\phi(x_{ij}), g_\phi(x_{ij})). \qquad (7)$$

$A$ is learned using its maximum a posteriori (MAP) estimate $\tilde{A}$, which is equivalent to using a Dirac measure as the approximated posterior distribution, $q_{\tilde{A}}(A) = \delta(A = \tilde{A})$, in the variational Bayes framework. The whole inference model is shown in Figure 1 (b). Therefore, $\tilde{A}, \theta, \phi$ are optimized by maximizing a lower-bound of the likelihood, i.e., evidence lower-bound (ELBO):

$$
\begin{aligned}
&\log p(X) \\
&= \log \int \frac{p_\theta(X|Z,A)P(A)P(Z)}{q_\phi(Z|X,A)} q_\phi(Z|X,A)dZdA \\
&\geq \mathbb{E}_{q_\phi(Z|X,\tilde{A})} \left[ \log p_\theta(X|Z,\tilde{A}) \right] + \log p(\tilde{A}) \quad (8) \\
&- \mathrm{KL} \left[ q_\phi(Z|X,\tilde{A})|p(Z) \right] \\
&= \mathcal{L}(\tilde{A}, \theta, \phi),
\end{aligned}
$$

where the first term is the expected reconstruction error of the gene expression matrix $X$, $\log p(\tilde{A})$ regularizes the MAP estimate of $A$, and $\mathrm{KL}\left[ q_\phi(Z|X,\tilde{A})|p(Z) \right]$ is the Kullback–Leibler divergence that regularizes the approximated posterior of $Z$ and is weighted by $\beta$ in practice.

In following subsections, we introduce our model InfoSEM by proposing flexible and informative priors for $p(A)$ to replace the weakly informative Laplace prior in Eq.5. Specifically, InfoSEM models the interaction effects, guided by textual gene embeddings, and the probability of interaction, informed by known interactions, when available.

### 3.2. Incorporate gene embeddings from pretrained language models

Given the frequent unavailability of costly experimental readouts, such as chromatin accessibility, RNA velocity, perturbation experiments, and accurate pseudo-time annotations, we propose the use of readily available priors. One such prior is gene embeddings derived from textual gene descriptions, which have been successfully utilized in recent supervised learning frameworks (Wang et al., 2024; Kommu et al., 2024). Gene-level information can assist GRN inference: if gene $i$ and gene $k$ are functionally similar, they may also have similar interaction effects with other genes. We use the $d$ dimensional embedding $\boldsymbol{h}_i \in \mathbb{R}^{1 \times d}$ of the textual description of the gene name from BioBERT (Lee et al., 2020), a language model pretrained on extensive biomedical literature, to represent the function of gene $i$. We design a prior of interaction effect informed by language model embeddings as shown in Figure 1 (c):

$$
\begin{aligned}
p(A|H, \boldsymbol{w}) &= \prod_{i,k} p(a_{ik}|\boldsymbol{h}_i, \boldsymbol{h}_k, \boldsymbol{w}) \\
&= \prod_{i,k} \mathrm{Laplace}(a_{ik}; [\boldsymbol{h}_i, \boldsymbol{h}_k]\boldsymbol{w}^T, \sigma_a),
\end{aligned} \quad (9)
$$

where $[\boldsymbol{h}_i, \boldsymbol{h}_k]$ concatenates the gene embedding of genes $i$ and $k$ into a $2d$-dim row vector and $\boldsymbol{w} \in \mathbb{R}^{1 \times 2d}$ is the parameter to learn using a MAP estimate with a prior $p(\boldsymbol{w}) = \mathcal{N}(0, \sigma_{\boldsymbol{w}}^2)$. The prior mean in Eq.9 can be interpreted as a linear model built on the textual gene embeddings for predicting their interactions. Although more flexible nonlinear models, e.g., MLP, can be applied, linear models on pre-trained embeddings have already been successful in several gene-level tasks (Chen & Zou, 2024).

The prior distribution defined in Eq.9 makes similar genes have similar interaction effects with other genes. Intuitively, if gene 2 and gene 3 are functionally close, i.e., $\boldsymbol{h}_2 \approx \boldsymbol{h}_3$, the prior defines a similar high-density region for $a_{12}$ and $a_{13}$ as they have a similar prior mean: $[\boldsymbol{h}_1, \boldsymbol{h}_2]\boldsymbol{w}^T \approx [\boldsymbol{h}_1, \boldsymbol{h}_3]\boldsymbol{w}^T$. Moreover, the prior is asymmetric, $p(a_{ik}|\boldsymbol{w}) \neq p(a_{ki}|\boldsymbol{w})$, which reflects the asymmetric nature of GRN.

From here on, we refer to this model, which uses BioBERT embeddings as priors, as InfoSEM-B.

### 3.3. Incorporate both gene embeddings from pretrained language models and known gene-gene interactions

Although not always readily available, prior knowledge of gene-gene interactions can be obtained in specific scenarios, such as those studied in (Yuan & Bar-Joseph, 2019; Anonymous, 2024; Chen & Liu, 2022), where ChIP-seq experiments or similar methodologies provide ground truth interaction data for subsets of $Y$. When available, these partially observed interactions offer valuable biological insights that can guide GRN inference. Here, we show how to incorporate these known interactions in addition to the textual gene embeddings previously described.

Incorporating known gene-gene interactions comes with a natural challenge in that the partially observed $Y$ is binary which does not inform the continuous weighted adjacency matrix $A$ directly, but it can inform the **probability** of interactions. Therefore, we propose to decompose the weighted adjacency matrix $A$ into $A^e \in \mathbb{R}^{P \times P}$, representing the magnitude of the interaction effect, and $A^l \in \mathbb{R}^{P \times P}$, with each element $a_{ik}^l$ representing the **logit of the probability** that gene $i$ interacts with gene $k$:

$$
A = A^e \odot \sigma(A^l), \quad (10)
$$

where $\sigma(\cdot)$ is the sigmoid function and $\odot$ is element-wise product (Hadamard product) between them, as shown in Figure 1 (d). We work on the logit space of probability to remove the necessary constraints during the model training. The prior of $A^e$ is informed by the gene embeddings from pretrained language models in the same way as Eq.9:

$$
p_e(A^e|H, \boldsymbol{w}) = \prod_{i,k} \mathrm{Laplace}(a_{ik}^e; [\boldsymbol{h}_i, \boldsymbol{h}_k]\boldsymbol{w}^T, \sigma_a). \quad (11)
$$

We define a prior on $A^l$ using the partially observed $Y$ as

following:

$$p_l(A^l|Y) = \prod_{i,k} p_l(a_{ik}^l|y_{ik}), \tag{12}$$

where

$$p_l(a_{ik}^l|y_{ik}) = \begin{cases} \mathcal{N}(\text{logit}(0.95), \sigma_l^2), & \text{if } y_{ik} = 1, \\ \mathcal{N}(\text{logit}(0.05), \sigma_l^2), & \text{if } y_{ik} = 0, \\ \boldsymbol{U}[\infty, \infty], & \text{if unknown}, \end{cases} \tag{13}$$

Intuitively, Eq.13 sets the mode of the prior probability that gene $i$ interacts with gene $k$ to 0.95 if we know they interact and 0.05 if they do not. The hyper-parameter $\sigma_l$ controls the precision of the labels. We truncate the binary label to 0.95 and 0.05 to ensure numerical stability (Skok Gibbs et al., 2024). If $y_{ik}$ is not observed, we use a non-informative uniform distribution $\boldsymbol{U}[\infty, \infty]$ as the prior of the logit, representing that the mode of the prior probability is 0.5.

We will henceforth refer to this model, which incorporates BioBERT embeddings as well as known interactions (from e.g., ChIP-seq) when available as priors, as InfoSEM-BC.

### 3.4. Learn InfoSEM with variational Bayes

We train InfoSEM-B (Section 3.2) and InfoSEM-BC (Section 3.3) using variational inference.

We use MAP estimates to infer all variables related to the weighted adjacency matrix, i.e., $\boldsymbol{w}$ in both models, $A$ in InfoSEM-B, and $A^e$ and $A^l$ in InfoSEM-BC, given the data $X$. We use a full-rank matrix $\tilde{A}, \tilde{A}^e \in \mathbb{R}^{P \times P}$ for $A$ and $A^e$, i.e., $q_{\tilde{A}}(A) = \delta(A = \tilde{A})$ and $q_{\tilde{A}^e}(A^e) = \delta(A^e = \tilde{A}^e)$. We use a low-rank MAP estimate with rank $h \ll P$ to infer $A^l$, motivated by the fact that the maximum rank of the adjacency matrix of a GRN is the number of transcription factors, as it represents the interactions from transcription factors to target genes (Li et al., 2020; Weighill et al., 2021). Therefore, we use $q_{A_a^l, A_b^l}(A^l) = \delta(A^l = A_a^l A_b^l)$, where $A_a^l \in \mathbb{R}^{P \times h}$, $A_b^l \in \mathbb{R}^{h \times P}$, and $A_a^l A_b^l$ has a rank $h$.

We derive the ELBO of InfoSEM-B with above model to be

$$\begin{aligned} &\mathcal{L}_{\text{InfoSEM-B}}(\tilde{A}, \theta, \phi, \boldsymbol{w}) \\ &= \mathbb{E}_{q_\phi(Z|X,\tilde{A})}\left[\log p_\theta(X|Z, \tilde{A})\right] + \log p(\tilde{A}|H, \boldsymbol{w}) \\ &+ \log p_w(\boldsymbol{w}) - \text{KL}\left[q_\phi(Z|X, \tilde{A})|p(Z)\right], \end{aligned} \tag{14}$$

and the ELBO of InfoSEM-BC to be

$$\begin{aligned} &\mathcal{L}_{\text{InfoSEM-BC}}(\tilde{A}^e, A_a^l, A_b^l, \theta, \phi, \boldsymbol{w}) \\ &= \mathbb{E}_{q_\phi(Z|X,\tilde{A}^e \odot \sigma(A_a^l A_b^l))}\left[\log p_\theta(X|Z, \tilde{A}^e \odot \sigma(A_a^l A_b^l))\right] \\ &+ \log p_e(\tilde{A}^e|H, \boldsymbol{w}) + \log p_l(A_a^l A_b^l|Y) + \log p_w(\boldsymbol{w}) \\ &- \text{KL}\left[q_\phi(Z|X, \tilde{A}^e \odot \sigma(A_a^l A_b^l))|p(Z)\right]. \end{aligned} \tag{15}$$

We provide the detailed derivation in the Appendix A.

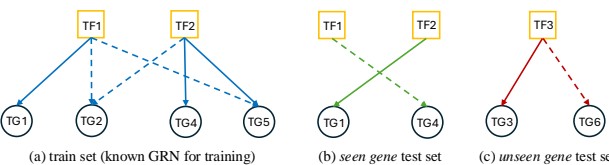

(a) train set (known GRN for training)  (b) *seen gene* test set  (c) *unseen gene* test set

*Figure 2.* Strategies of train-test splits of the ground-truth network, containing three transcription factors (TF1-TF3) and six target genes (TG1-TG6), when it is used during training. Solid and dashed lines represent positive and negative interactions. (a) Blue links ($\rightarrow$) represent known interactions used in the training set. (b) Green links ($\rightarrow$) represent the test set $\mathcal{D}_{\text{seen}}^{\text{test}}$, i.e., pairs of genes for which the individual genes are seen in the train data, but not their interaction. (c) Red links ($\rightarrow$) represent the test set $\mathcal{D}_{\text{unseen}}^{\text{test}}$, i.e., pairs that both the genes and their interactions are unseen in the train data.

## 4. A new benchmarking framework for evaluating GRNs

Evaluation methodologies for GRN inference models play a critical role in validating their biological relevance and utility. Existing GRN evaluation benchmarks, while suitable for specific tasks, have limitations. We highlight these shortcomings first and propose a biologically motivated framework to address them.

### 4.1. Limitations of existing supervised learning benchmarks

When using parts of the ground-truth network to train a GRN inference model, careful train-test split strategies are essential for reliable evaluation. In most supervised GRN inference studies (Chen & Liu, 2022; Wang et al., 2023; Mao et al., 2023; Xu et al., 2023; Wang et al., 2024; Kommu et al., 2024), datasets are split by **gene pairs**, where edges for each TF are randomly divided into training and test sets. This results in a test set, $\mathcal{D}_{\text{seen}}^{\text{test}}$, containing *unseen interactions between seen genes*. For instance, in Figure 2 (b), the link from TF2 to TG1 is in $\mathcal{D}_{\text{seen}}^{\text{test}}$, unseen during training, though both TF2 and TG1 appear in the training set (Figure 2 (a)). This means that TFs and TGs in test sets are also in training sets, with only the regulatory interactions (edges) being distinct between the training and test sets. Furthermore, the class imbalance for interactions associated with each TF remains consistent across $\mathcal{D}^{\text{train}}$ and $\mathcal{D}_{\text{seen}}^{\text{test}}$. This approach has the following major limitations, which are further explored in Section 5:

- Risk of memorization due to class imbalance: The partially observed ground-truth GRN networks are often heavily imbalanced (Anonymous, 2024; Pratapa et al., 2020): most TFs have far more negative interactions (no regulation) than positive ones (Figure 3 (b) right). When all genes are represented in both training and test sets,

| Methods | scRNA-seq | Known GRN | External prior | Framework |
|---|---|---|---|---|
| One-hot LR | ✗ | ✓ | ✗ | SL |
| Matrix Completion | ✗ | ✓ | ✗ | SL |
| scGREAT | ✓ | ✓ | ✓ | SL |
| GENELink | ✓ | ✓ | ✗ | SL |
| GRNBoost2 | ✓ | ✗ | ✗ | USL |
| DeepSEM | ✓ | ✗ | ✗ | USL |
| InfoSEM-B (Ours) | ✓ | ✗ | ✓ | USL |
| InfoSEM-BC (Ours) | ✓ | ✓ | ✓ | USL |

*Table 1.* Properties of a list of benchmarking methods. Existing methods that make use of known GRN interactions treat them as prediction targets in supervised learning (SL) framework while our approaches consider them as a prior for constructing the scRNA-seq in an unsupervised learning (USL) framework.

models can memorize gene-specific patterns (e.g., gene IDs) by exploiting their node degrees from the GT network rather than learning true regulatory mechanisms using the gene expression data. Such memorization undermines the biological validity of the inferred networks.

• Inadequate representation of real-world scenarios: In real-world cases such as biomarker expansion studies, the interaction (degree) information of most genes is not represented in the partially observed GRN (Lotfi Shahreza et al., 2018). For instance, TF3, TG3, and TG6 in Figure 2 (c) exhibit this scenario. Evaluating models on *unseen genes* (genes not present in the training set) reflects their ability to generalize to new biological contexts. This scenario is not captured by existing benchmarks.

### 4.2. Proposed benchmarking framework

In this work, we construct another test set, $\mathcal{D}_{\text{unseen}}^{\text{test}}$, containing *interaction between unseen genes*, i.e., all **genes** in $\mathcal{D}_{\text{unseen}}^{\text{test}}$ are not in the training set (see Figure 2 (c) for an example). The model performance and generalization ability on $\mathcal{D}_{\text{unseen}}^{\text{test}}$ would reflect the level of biology that the model has learned from the gene expression data. In practice, we randomly divide all TFs and TGs with known interactions into four sets: seen and unseen TFs, e.g., [TF1, TF2] and [TF3] in Figure 2, and seen and unseen TGs, e.g., [TG1, TG2, TG4, TG5] and [TG3, TG6], with a ratio 3:1. All links between unseen TFs and unseen TGs are in $\mathcal{D}_{\text{unseen}}^{\text{test}}$ (e.g., TF3 → TG3). We then randomly divide all interactions between seen TFs and seen TGs into training $\mathcal{D}^{\text{train}}$ and $\mathcal{D}_{\text{seen}}^{\text{test}}$ with a ratio 3:1 using split strategies in existing works. Essentially, we leave some TFs and TGs out when constructing the training set whose interactions are then considered as the unseen gene test sets $\mathcal{D}_{\text{unseen}}^{\text{test}}$.

## 5. Experiments

We begin by experimentally validating the limitations of existing GRN inference benchmarks mentioned in the previous subsection to motivate the need for our new benchmarking

setup followed by comparison of our proposed InfoSEM model with state-of-the-art supervised and unsupervised models. For reproducibility, all experimental details are available in Appendix D.

For all experiments, we use scRNA-seq datasets of four cell lines from the popular BEELINE suite (Pratapa et al., 2020), including human embryonic stem cells (hESC) (Yuan & Bar-Joseph, 2019), human mature hepatocytes (hHEP) (Camp et al., 2017), mouse dendritic cells (mDC) (Shalek et al., 2014), and mouse embryonic stem cells (mESC) (Hayashi et al., 2018). We consider two available ground-truth networks, cell-type specific ChIP-seq, collected from databases such as ENCODE and ChIP-Atlas, on the same or similar cell type, and non-cell-type specific transcriptional regulatory network from BEELINE (Pratapa et al., 2020).

We compare nine methods, including: 1. a random baseline; 2. two trivial methods that do not use any information from scRNA-seq: one-hot LR (a logistic regression that takes the concatenation of the one-hot embedding of two genes to predict the ground-truth interactions) and matrix completion (MatComp) that imputes the partially observed adjacency matrix $Y$ using low-rank decomposition (Troyanskaya et al., 2001); 3. two recent supervised methods with state-of-the-art reported performances: scGREAT (Wang et al., 2024) and GENELink (Chen & Liu, 2022); 4. two state-of-the-art unsupervised methods: GRNBoost2 (Moerman et al., 2019) and DeepSEM (Shu et al., 2021); 5. our proposed InfoSEM-B and InfoSEM-BC models. Inputs and training objectives for all models are summarized in Table 1.

We consider two types of evaluation metrics: accuracy-based metric: AUPRC (Yang et al., 2022), and rank-based metric (Pratapa et al., 2020): number of positive interactions among the top 1% (Hit@1%) predicted interactions. We repeat the train-test splits 10 times with different random seeds to estimate the error bar for each method.

### 5.1. Exploring the biases with existing benchmarks

Under the existing benchmarks involving *unseen interactions between seen genes*, we observe that the performance of existing supervised and unsupervised methods aligns with prior studies (Wang et al., 2024; Chen & Liu, 2022; Pratapa et al., 2020). However, no previous work has compared these methods to simple supervised baselines, one-hot LR, which uses only one-hot embeddings of gene IDs as features, and a matrix completion baseline, which simply imputes the partially observed $Y$ without any additional information. None of these use any information from scRNA-seq data. Surprisingly, both one-hot LR and matrix completion perform similarly to state-of-the-art supervised models such as scGREAT and GENELink across all cell lines (Figure 3 (a) first column for hESC and hHEP cell lines, full results in Appendix F.1 for all cell lines show similar trend).

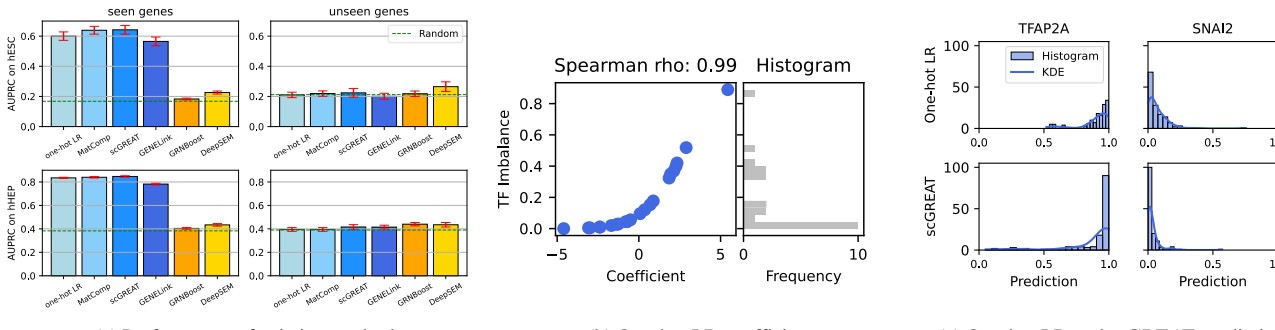

(a) Performance of existing methods      (b) One-hot LR coefficients      (c) One-hot LR and scGREAT predictions

*Figure 3.* (a) AUPRC with corresponding standard error of the mean of **existing** GRN inference models with **cell-type specific** target on both *unseen interactions between seen genes* (first column) and *interactions between unseen genes* (second column) test sets. One-hot LR and matrix completion without gene expression data achieve similar performance as several latest supervised methods (scGREAT, GENELink) using gene expression. (b) Relations between class imbalance level and one-hot LR coefficient of each TF. The x-axis shows the coefficient of one-hot embedding in LR. The y-axis represents the imbalance level (proportion of positive interactions) of each TF with histogram shown on the right. (c) Histograms and kernel density estimations of predicted probabilities from one-hot LR and scGREAT on interactions in $\mathcal{D}_{\text{seen}}^{\text{test}}$ with two TFs: TFAP2A (left) and SNAI2 (right), with training class imbalance level 0.89 and 0.06 respectively.

This performance is driven by one-hot LR's ability to exploit the node degree from the partially known GRN. Specifically, the one-hot embeddings enable the model to memorize gene-specific biases, such as the probability of a transcription factor (TF) regulating a target gene (TG). To validate this, we analyzed the correlation between the class imbalance level (the ratio of positive to negative interactions) of each TF and the corresponding one-hot coefficient in the one-hot LR. Figure 3 (b) shows a strong rank correlation, supporting that one-hot LR predicts interactions based on dataset biases to match the class imbalance level of the corresponding TF, rather than biological relationships. Similarly, Mat-Comp works well by learning degree biases in the partially observed network of genes with low-rank latent features, which generalizes effectively to unseen interactions between those seen genes in the partially observed network.

In imbalanced datasets, where certain genes are predominantly associated with positive or negative interactions, supervised models can achieve high performance by learning these biases, rather than meaningful biological relationships from gene expression data. The cross-entropy loss allows supervised models (Eq.1) to exploit this class imbalance by learning, for example, how likely a TF is to regulate TGs, or how likely a TG is to be regulated, without relying on gene expression data. This shortcut allows models to predict *unseen interactions between seen genes* based solely on these biases. For illustration, both one-hot LR and scGREAT predict unseen interactions of a seen TF (e.g., TFAP2A in Figure 3 (c)) positive if the TF has more positive interactions in the training set and vice versa (e.g., SNAI2).

To systematically confirm further whether sophisticated supervised models, such as scGREAT and GENELink, also rely on dataset biases, we evaluated them on our new benchmark designed to predict *interactions between unseen genes*

using scRNA-seq data. In this scenario, we observe a significant performance drop for supervised GRN inference methods, averaging 42% for cell-specific ChIP-seq and 79% for non-cell-specific ChIP-seq (see Figure 3 (a) second column, Table 2, and Appendix Table 6), even underperforming the random baseline (dashed green line in Figure 3 (a)). In contrast, unsupervised methods, especially DeepSEM, outperformed supervised methods on three datasets (hESC, hHEP, mDC) for cell-specific ChIP-seq without using known interactions. This further confirms the dependence of supervised methods on dataset-specific biases. Using simple techniques such as downsampling to address class imbalance in supervised methods reduces their accuracy inflation on existing benchmarks but does not improve their accuracy on *unseen genes* scenario (see Appendix E for details). This trend holds across both cell-type-specific and non-cell-type-specific benchmarks, and for metrics such as Hit@1% (additional results in the Appendix F).

### 5.2. Utility of informative priors for InfoSEM

Next, we study the performance of our proposed InfoSEM models on our introduced *unseen genes* benchmark. Table 2 and Appendix Table 6 demonstrate that InfoSEM-BC and InfoSEM-B emerge as the best-performing models on all datasets that we tested for both cell-specific and non-cell-specific ChIP-seq. InfoSEM-B, which does not use known interactions, always improves upon DeepSEM on all datasets that we tested for both cell-specific and non-cell-specific cases by 25% and 52% on average, respectively. InfoSEM-B, without using any known interaction labels, achieves top-2 performance for both AUPRC and Hit@1% metrics on 3 out of 4 datasets, even when compared to SL baselines that incorporate known interactions (see Table 2 and Appendix Tables 6 and 7).

| | hESC | | hHEP | | mDC | | mESC | |
|---|---|---|---|---|---|---|---|---|
| | AUPRC | Hit@1% | AUPRC | Hit@1% | AUPRC | Hit@1% | AUPRC | Hit@1% |
| One-hot LR | 0.210 (0.018) | 0.205 (0.041) | 0.395 (0.016) | 0.345 (0.056) | 0.247 (0.019) | 0.225 (0.104) | 0.329 (0.026) | 0.397 (0.036) |
| MatComp | 0.219 (0.018) | 0.191 (0.048) | 0.395 (0.016) | 0.356 (0.034) | 0.240 (0.008) | 0.225 (0.043) | 0.342 (0.023) | 0.345 (0.038) |
| scGREAT | 0.224 (0.029) | 0.222 (0.046) | 0.416 (0.020) | 0.433 (0.047) | 0.245 (0.017) | 0.183 (0.035) | **0.393 (0.027)** | 0.390 (0.066) |
| GENELink | 0.201 (0.020) | 0.144 (0.050) | 0.415 (0.016) | 0.428 (0.060) | 0.249 (0.013) | 0.207 (0.054) | 0.381 (0.030) | 0.454 (0.094) |
| SCENIC | 0.210 (0.020) | 0.200 (0.037) | 0.465 (0.020) | **0.568 (0.047)** | 0.227 (0.014) | 0.219 (0.062) | 0.346 (0.024) | 0.393 (0.045) |
| GRNBoost2 | 0.218 (0.018) | 0.162 (0.037) | 0.440 (0.014) | **0.553 (0.058)** | 0.226 (0.009) | 0.167 (0.053) | 0.356 (0.023) | 0.348 (0.049) |
| DeepSEM | 0.265 (0.032) | 0.419 (0.059) | 0.435 (0.019) | 0.517 (0.043) | 0.277 (0.014) | 0.292 (0.095) | 0.343 (0.024) | 0.369 (0.048) |
| InfoSEM-B (Ours) | **0.331 (0.055)** | **0.547 (0.091)** | **0.498 (0.020)** | 0.533 (0.048) | **0.298 (0.028)** | **0.472 (0.076)** | 0.388 (0.023) | **0.522 (0.047)** |
| InfoSEM-BC (Ours) | **0.331 (0.056)** | **0.585 (0.094)** | **0.499 (0.020)** | 0.550 (0.053) | **0.322 (0.032)** | **0.498 (0.069)** | **0.408 (0.020)** | **0.575 (0.045)** |
| Random | 0.222 | 0.215 | 0.390 | 0.397 | 0.232 | 0.231 | 0.349 | 0.351 |

*Table 2.* The GRN inference performance, measured by AUPRC and Hit@1%, with corresponding standard error of the mean (in parentheses) for each method evaluated on *interactions between unseen genes* with **cell-type specific ChIP-seq** targets. The top-2 best models are **bold**.

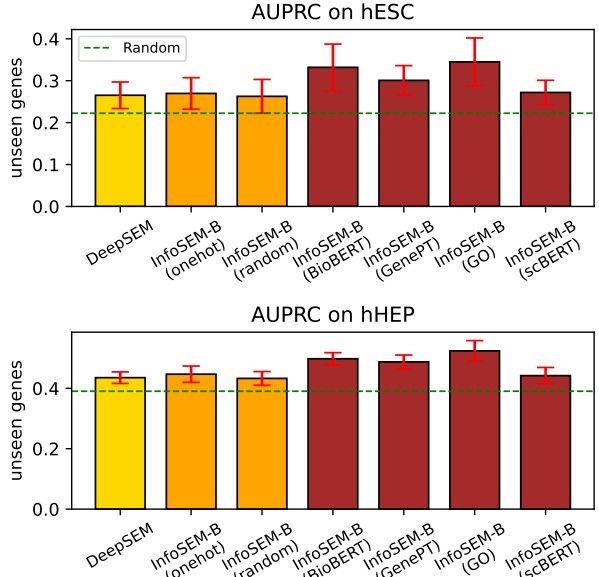

*Figure 4.* Results of using different prior gene embeddings in InfoSEM-B with **cell-type specific ChIP-seq** targets. InfoSEM-B with informative gene embedding, e.g., from BioBERT, GenePT, and gene ontology (GO), are better than models with noninformative embeddings, e.g., one-hot and random. However, InfoSEM-B with gene embeddings from scBERT does not show significant improvement.

InfoSEM-BC which incorporates information from known interactions achieves the best performance on all datasets with respect to AUPRC and on 3 out of 4 datasets with respect to Hit@1% for both cell-specific and non-cell-specific ChIP-seq target on *unseen genes* test set with average improvements over DeepSEM by 31% and 66% (see Table 2 and Appendix Tables 6 and 7). By integrating prior knowledge from known interactions, InfoSEM-BC reconstructs scRNA-seq data effectively, enabling strong performance on *unseen genes* without the pitfalls of class imbalance exploited by supervised methods relying on their discrimi-

native loss (Eq.1).

**Run-time analysis:** We include comparison of run times between InfoSEM and other baselines on different datasets with various numbers of cells and genes in Appendix B. Our results demonstrate that InfoSEM is faster than popular supervised learning baselines and only slightly slower than DeepSEM, while consistently outperforming them in terms of performance in unseen genes benchmark.

**Sensitivity analysis:** Since InfoSEM-B uses embeddings from BioBERT, it is natural to ask: how useful are these embeddings in improving GRN inference? To explore this, we investigate the impact of alternative gene embeddings, including textual gene embeddings from another language model GenePT (Chen & Zou, 2024), gene embeddings derived from gene ontology (GO) (Ashburner et al., 2000), and gene embeddings from a single cell foundation model scBERT (Yang et al., 2022). Figure 4 shows that InfoSEM-B models replace BioBERT embeddings with other gene embeddings, such as InfoSEM-B (GenePT) and InfoSEM-B (GO), still resulting in improved GRN inference compared to DeepSEM. As expected, using non-informative embeddings, such as one-hot or random embeddings, in InfoSEM-B does not enhance performance. Interestingly, we observe using embeddings from scBERT does not improve performance. We hypothesize that this may result from the binning of scRNA-seq data in scBERT, which could lead to a loss of important information. Additionally, embeddings from scBERT, derived from binned scRNA-seq data, are similar to those used in InfoSEM already and may not provide complementary insights for the adjacency matrix compared to embeddings like gene ontology or BioBERT, which are derived independent of scRNA-seq data. We believe, though, that exploring other scRNA-seq foundation model derived gene embeddings presents an interesting avenue for future exploration.

Additionally, we evaluate the sensitivity of GRN inference accuracy with respect to the number of cells in the train-

ing data, as detailed in Appendix F.3. We find that both InfoSEM-B and InfoSEM-BC, trained on just 20% of cells, achieve performance comparable to DeepSEM trained on all cells for the hESC cell line.

## 6. Conclusion and discussion

In this work, we study the problem of Gene Regulatory Network (GRN) inference using scRNA-seq data. Existing benchmarks focus on interactions between *seen genes*, i.e., all genes appear in both training and test sets. While suitable for predicting novel interactions within well-characterized gene sets, they fail to address real-world applications such as biomarker expansion (Lotfi Shahreza et al., 2018), where models must generalize to interactions between unseen genes with no prior knowledge about their interactions. Additionally, we show that the high performance of supervised methods even on existing benchmarks may be influenced by dataset (gene-specific biases, such as class imbalance, rather than their ability to learn true biological mechanisms.

To address this gap, we propose a new, biologically motivated benchmarking framework that evaluates a model's ability to infer interactions between unseen genes. We also introduce InfoSEM, an unsupervised generative model that integrates biologically meaningful priors by leveraging textual gene embeddings from BioBERT. InfoSEM achieves 38.5% on average improvement over existing state-of-the-art models without informative priors for GRN inference. Furthermore, we show that InfoSEM can incorporate known interaction labels when available, further enhancing performance by 11.1% on average across datasets while avoiding the pitfalls of training with class imbalance.

We believe InfoSEM's ability to generalize to unseen gene interactions, even with no or minimal labeled data, makes it a powerful tool for real-world applications, such as target discovery/prioritization. It can also serve as a key component in active learning frameworks, guiding the acquisition of new interactions from wet-lab experiments and expanding our understanding of GRNs in practical scenarios.

## Impact Statement

Understanding gene regulatory networks (GRNs) is fundamental to deciphering cellular processes, with broad applications in disease research, drug discovery, and biomarker identification. Our work contributes to this field by introducing a more rigorous evaluation framework for GRN inference, emphasizing generalization to previously unseen genes—a critical challenge in real-world biological studies.

From an ethical standpoint, advances in GRN inference carry both opportunities and risks. Improved inference methods can accelerate biomedical research, leading to more precise diagnostics and targeted therapies. However, reliance on biases in existing datasets and benchmarks datasets may introduce artifacts that misrepresent true regulatory interactions, potentially leading to incorrect conclusions in downstream analyses. Transparency in model evaluation and careful benchmarking, as emphasized in our work, are essential to mitigate such risks.

Additionally, as GRN models improve in predictive accuracy, their impact on biomedical research and healthcare decisions will grow. It is crucial to ensure that these models are evaluated rigorously and deployed responsibly, avoiding over-interpretation of predictions and ensuring their applicability across diverse biological contexts. Ethical considerations should guide their use, promoting transparency, reproducibility, and equitable advancements in genomics and medicine.

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

# A. Derivation of ELBO for InfoSEM-B and InfoSEM-BC.

In this section, we derive the evidence lower bound (ELBO) of InfoSEM with two proposed priors.

## A.1. InfoSEM-B

$$
\begin{aligned}
&p(X|H) \\
&= \int p_\theta(X|Z, A)p(A|H, \boldsymbol{w})p_w(\boldsymbol{w})p(Z)dZdA \\
&= \log \int \frac{p_\theta(X|Z, A)p(A|H, \boldsymbol{w})p_w(\boldsymbol{w})p(Z)}{q_\phi(Z|X, A)}q_\phi(Z|X, A)dZdA \\
&= \log \int \frac{p_\theta(X|Z, \tilde{A})p(\tilde{A}|H, \boldsymbol{w})p_w(\boldsymbol{w})p(Z)}{q_\phi(Z|X, \tilde{A})}q_\phi(Z|X, \tilde{A})dZ \\
&\geq \mathbb{E}_{q_\phi(Z|X,\tilde{A})}\left[\log p_\theta(X|Z, \tilde{A})\right] + \log p(\tilde{A}|H, \boldsymbol{w}) + \log p_w(\boldsymbol{w}) - \mathrm{KL}\left[q_\phi(Z|X, \tilde{A})|p(Z)\right] \\
&= \mathcal{L}_{\text{InfoSEM-B}}(\tilde{A}, \theta, \phi, \boldsymbol{w}).
\end{aligned}
\tag{16}
$$

## A.2. InfoSEM-BC

$$
\begin{aligned}
&p(X|H, Y) \\
&= \int p_\theta(X|Z, A^e \odot \sigma(A^l))p_e(A^e|H, \boldsymbol{w})p_l(A^l|Y)p_w(\boldsymbol{w})p(Z)dZdA^edA^l \\
&= \log \int \frac{p_\theta(X|Z, A^e \odot \sigma(A^l))p_e(A^e|H, \boldsymbol{w})p_l(A^l|Y)p_w(\boldsymbol{w})p(Z)}{q_\phi(Z|X, A^e \odot \sigma(A^l))}q_\phi(Z|X, A^e \odot \sigma(A^l))dZdA^edA^l \\
&= \log \int \frac{p_\theta(X|Z, \tilde{A}^e \odot \sigma(A_a^lA_b^l))p_e(\tilde{A}^e|H, \boldsymbol{w})p_l(A_a^lA_b^l|Y)p_w(\boldsymbol{w})p(Z)}{q_\phi(Z|X, \tilde{A}^e \odot \sigma(A_a^lA_b^l))}q_\phi(Z|X, \tilde{A}^e \odot \sigma(A_a^lA_b^l))dZ \\
&\geq \mathbb{E}_{q_\phi(Z|X,\tilde{A}^e\odot\sigma(A_a^lA_b^l))}\left[p_\theta(X|Z, \tilde{A}^e \odot \sigma(A_a^lA_b^l))\right] + \log p_e(\tilde{A}^e|H, \boldsymbol{w}) + \log p_l(A_a^lA_b^l|Y) + \log p_w(\boldsymbol{w}) \\
&- \mathrm{KL}\left[q_\phi(Z|X, \tilde{A}^e \odot \sigma(A_a^lA_b^l))|p(Z)\right] \\
&= \mathcal{L}_{\text{InfoSEM-BC}}(\tilde{A}^e, A_a^l, A_b^l, \theta, \phi, \boldsymbol{w}).
\end{aligned}
\tag{17}
$$

# B. Run-time analysis

| (#cell, #gene) | scGREAT | GENELink | SCENIC | GRNBoost2 | DeepSEM | InfoSEM-B | InfoSEM-BC |
|---|---|---|---|---|---|---|---|
| (454, 844) | 236.30 | 177.25 | 80.81 | 9.27 | 91.88 | 118.00 | 125.91 |
| (758, 844) | 239.96 | 172.72 | 97.74 | 10.81 | 112.55 | 134.11 | 164.16 |
| (758, 1291) | 357.52 | 258.91 | 142.51 | 13.74 | 116.01 | 206.86 | 211.37 |

*Table 3.* Run-time analysis

We include comparison of run times (in seconds) for different configurations of the datasets in terms of number of cells and genes, (#cell, #gene), for InfoSEM and other baselines in Table 3. Our results demonstrate that InfoSEM is faster than popular supervised learning baselines and only slightly slower than DeepSEM, while consistently outperforming them in terms of performance in unseen genes benchmark.

# C. Details of each dataset

## C.1. Cell-type specific datasets

|  | hESC | hHEP | mDC | mESC |
|---|---|---|---|---|
| number of genes | 844 | 908 | 1216 | 1353 |
| number of cells | 758 | 425 | 383 | 421 |
| number of positive links | 4404 | 9684 | 1129 | 40083 |
| number of negative links | 23448 | 17556 | 24407 | 78981 |
| averaged node degree of TFs | 133.5 | 322.8 | 53.8 | 455.5 |

## C.2. Non-cell-type specific datasets

|  | hESC | hHEP | mDC | mESC |
|---|---|---|---|---|
| number of genes | 844 | 908 | 1216 | 1353 |
| number of cells | 758 | 425 | 383 | 421 |
| number of positive links | 3318 | 4033 | 3694 | 7705 |
| number of negative links | 229626 | 279263 | 300306 | 678266 |
| averaged node degree of TFs | 12.0 | 12.9 | 14.8 | 15.2 |

# D. Reproducibility: details and hyperparameters of each model

In general, we cross-validate hyper-parameters using the partially known GRN on the training set to ensure a fair comparison with existing methods that use the same strategy. Code and datasets will be made available on acceptance.

**One-hot LR:** A logistic regression implemented by scikit-learn (Pedregosa et al., 2011) with the $L_2$ regularization coefficient cross-validated on the training set.

**MatComp:** A matrix completion algorithm implemented by fancyimpute (Rubinsteyn & Feldman) with rank 128.

**scGREAT:** Use hyperparameters and implementation provided by Wang et al. (2024).

**GENELink:** Use hyperparameters and implementation provided by Chen & Liu (2022).

**DeepSEM:** Use hyperparameters and implementation provided by Yuan & Bar-Joseph (2019).

**InfoSEM-B and InfoSEM-BC:** We set hyper-parameters, e.g., neural network architectures, learning rates schedule, prior scale of latent variable $Z$, to be the same as DeepSEM (Yuan & Bar-Joseph, 2019). For unique hyper-parameters of InfoSEM-B and InfoSEM-BC, we cross-validate them on the training set using known GRN (shown as below).

| Ground-truth | Methods | hESC | hHEP | mDC | mESC |
|---|---|---|---|---|---|
| cell-type specific ChIP-seq | $\sigma_{\boldsymbol{w}}$ (InfoSEM-B, InfoSEM-BC) | 0.1 | 10 | 10 | 1 |
|  | $\sigma_l$ (InfoSEM-BC) | 0.1 | $\sqrt{0.1}$ | $\sqrt{0.1}$ | 0.1 |
|  | $h$ (InfoSEM-BC) | 128 | 128 | 128 | 128 |
| non-cell-type specific ChIP-seq | $\sigma_{\boldsymbol{w}}$ (InfoSEM-B, InfoSEM-BC) | 100 | 100 | 100 | 100 |
|  | $\sigma_l$ (InfoSEM-BC) | $\sqrt{0.1}$ | 0.1 | $\sqrt{0.1}$ | $\sqrt{0.1}$ |
|  | $h$ (InfoSEM-BC) | 128 | 128 | 128 | 128 |

# E. Performance of supervised methods with downsampling

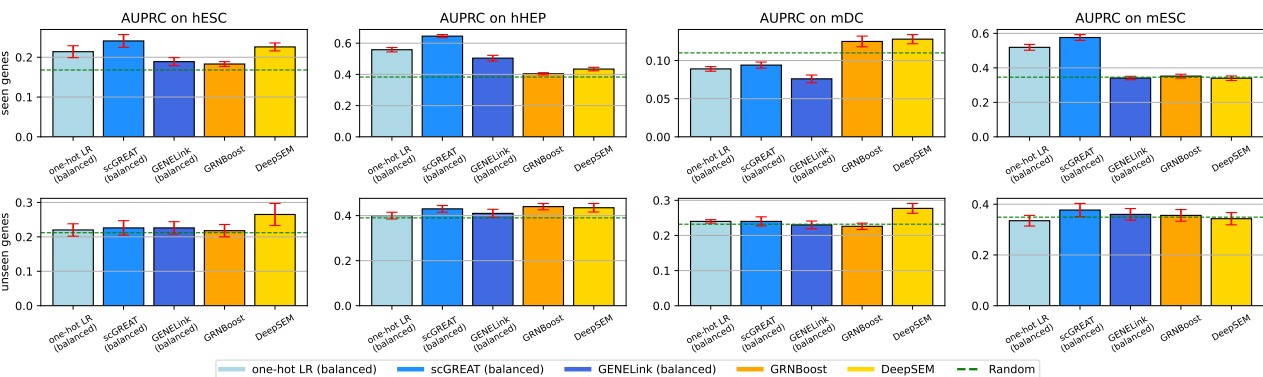

*Figure 5.* AUPRC with corresponding standard error of the mean of GRN inference models with **cell-type specific** target on *unseen interactions between seen genes* test sets. Although we apply downsampling to remove the class imbalance level associated with each TF, supervised methods still show an inflated accuracy on most of cell lines by using the class imbalance level associated with each TG.

If the class imbalance level associated with each gene causes the inflated accuracy of supervised method, one straightforward solution is to remove the class imbalance with downsampling. Specifically, we randomly select the same number of negative edges as the number of positive edges for each TF when training a supervised model, and the performance is shown in Fig 5.

We observe that supervised methods still have an inflated accuracy by comparing their performance on *unseen interactions between seen genes* and *interactions between unseen genes*. One reason is that edges associated with TGs can still be imbalanced even the edges associated with TFs are balanced, and supervised learning methods can still make use of such information easily.

Moreover, removing the class imbalance of TFs does not improve the performance on *interactions between unseen genes* and unsupervised methods, especially DeepSEM, are still outperform supervised methods on *unseen genes*.

# F. Additional experimental results

## F.1. Performance of all methods on *unseen interaction between seen genes* test set.

| | hESC | | hHEP | | mDC | | mESC | |
|---|---|---|---|---|---|---|---|---|
| | AUPRC | Hit@1% | AUPRC | Hit@1% | AUPRC | Hit@1% | AUPRC | Hit@1% |
| One-hot LR | **0.600 (0.028)** | 0.932 (0.039) | 0.835 (0.005) | **1.000 (0.003)** | 0.176 (0.006) | 0.250 (0.024) | 0.844 (0.009) | **0.991 (0.004)** |
| MatComp | 0.638 (0.026) | 0.913 (0.037) | **0.840 (0.006)** | **1.000 (0.006)** | **0.351 (0.013)** | **0.419 (0.034)** | **0.860 (0.007)** | 0.974 (0.008) |
| scGREAT | **0.642 (0.029)** | **0.966 (0.039)** | **0.847 (0.006)** | **1.000 (0.000)** | 0.249 (0.011) | 0.325 (0.047) | **0.858 (0.009)** | **1.000 (0.001)** |
| GENELink | 0.565 (0.029) | **0.964 (0.043)** | 0.782 (0.007) | 0.942 (0.015) | 0.178 (0.006) | 0.162 (0.026) | 0.690 (0.016) | 0.716 (0.101) |
| SCENIC | 0.177 (0.008) | 0.176 (0.031) | 0.396 (0.010) | 0.449 (0.031) | 0.119 (0.008) | 0.171 (0.044) | 0.328 (0.012) | 0.380 (0.026) |
| GRNBoost2 | 0.173 (0.006) | 0.230 (0.017) | 0.384 (0.007) | 0.423 (0.015) | 0.115 (0.007) | 0.141 (0.036) | 0.352 (0.011) | 0.342 (0.024) |
| DeepSEM | 0.216 (0.010) | 0.318 (0.016) | 0.424 (0.011) | 0.509 (0.028) | 0.118 (0.006) | 0.165 (0.033) | 0.340 (0.013) | 0.431 (0.023) |
| InfoSEM-B | 0.374 (0.027) | 0.752 (0.067) | 0.467 (0.011) | 0.538 (0.020) | 0.130 (0.010) | 0.174 (0.048) | 0.401 (0.012) | 0.562 (0.016) |
| InfoSEM-BC | 0.553 (0.028) | 0.913 (0.053) | 0.702 (0.007) | 0.865 (0.023) | **0.285 (0.016)** | **0.555 (0.063)** | 0.672 (0.010) | 0.918 (0.026) |
| Random | 0.168 | 0.171 | 0.383 | 0.384 | 0.110 | 0.111 | 0.346 | 0.344 |

*Table 4.* The GRN inference performance for each method evaluated on *unseen interactions between seen genes* with **cell-type specific ChIP-seq** targets. The top-2 best models are **bold**. We observe that all supervised learning methods, including one-hot LR, achieve much better performance than unsupervised methods.

| | hESC | | hHEP | | mDC | | mESC | |
|---|---|---|---|---|---|---|---|---|
| | AUPRC | Hit@1% | AUPRC | Hit@1% | AUPRC | Hit@1% | AUPRC | Hit@1% |
| One-hot LR | 0.159 (0.001) | 0.257 (0.013) | 0.186 (0.004) | 0.306 (0.008) | 0.126 (0.004) | 0.228 (0.007) | 0.127 (0.004) | 0.205 (0.006) |
| MatComp | **0.207 (0.017)** | **0.334 (0.020)** | **0.247 (0.012)** | **0.375 (0.016)** | **0.396 (0.009)** | **0.563 (0.015)** | **0.260 (0.009)** | **0.375 (0.013)** |
| scGREAT | **0.173 (0.002)** | 0.288 (0.014) | **0.244 (0.006)** | **0.366 (0.004)** | 0.183 (0.012) | 0.314 (0.004) | 0.139 (0.004) | 0.235 (0.004) |
| GENELink | 0.059 (0.011) | 0.101 (0.022) | 0.088 (0.011) | 0.157 (0.022) | 0.103 (0.007) | 0.141 (0.019) | 0.025 (0.005) | 0.025 (0.013) |
| SCENIC | 0.019 (0.001) | 0.033 (0.004) | 0.019 (0.001) | 0.036 (0.003) | 0.020 (0.001) | 0.036 (0.004) | 0.021 (0.001) | 0.044 (0.003) |
| GRNBoost2 | 0.019 (0.001) | 0.031 (0.004) | 0.018 (0.001) | 0.031 (0.003) | 0.020 (0.001) | 0.031 (0.005) | 0.018 (0.001) | 0.042 (0.003) |
| DeepSEM | 0.023 (0.001) | 0.034 (0.004) | 0.021 (0.001) | 0.035 (0.004) | 0.021 (0.001) | 0.033 (0.005) | 0.021 (0.001) | 0.044 (0.002) |
| InfoSEM-B | 0.025 (0.001) | 0.038 (0.005) | 0.021 (0.001) | 0.037 (0.002) | 0.022 (0.001) | 0.038 (0.003) | 0.023 (0.000) | 0.042 (0.002) |
| InfoSEM-BC | 0.119 (0.008) | **0.296 (0.018)** | 0.171 (0.008) | 0.331 (0.011) | **0.325 (0.010)** | **0.516 (0.016)** | **0.159 (0.006)** | **0.309 (0.007)** |
| Random | 0.019 | 0.017 | 0.018 | 0.017 | 0.018 | 0.018 | 0.014 | 0.014 |

*Table 5.* The GRN inference performance for each method evaluated on *unseen interactions between seen genes* with **non-cell-type specific ChIP-seq** targets. The top-2 best models are **bold**. We observe that InfoSEM-BC achieves top-2 performance on three dataset.

We show the performance of all methods evaluated under current evaluation setup, i.e., *unseen interactions between seen genes*, in Table 4 and Table 5 for cell-type specific and non-cell-type specific GRNs. We observe that trivial baselines (one-hot LR and matrix completion) without using any gene expression data always achieve top-2 performance.

## F.2. Performance of methods on *interaction between unseen genes* test set for non-cell-type specific ChIP-seq target.

We show the performance of all methods evaluated on *interactions between unseen genes* for the non-cell-type specific GRNs in Table 6, where the proposed InfoSEM is among top-2 best models in all cell lines. We observe that both AUPRC and Hit@1% are very small when benchmarked against non-cell-type specific GRNs unlike cell specific GRNs in Table 2. One reason is that negative links in non-cell-type specific GRNs contain both unknown positive links and negative links due to the data collection process (Pratapa et al., 2020), therefore, negative links in non-cell-type specific GRNs are much noisier than cell-type specific GRNs. However, positive links in non-cell-type specific GRNs do not contain such noise. We compute the recall to evaluate the performance of our methods on positive links only (Table 7) where we observe a much higher accuracy compared with Table 6 when both positive links and negative links are evaluated.

## F.3. Sensitivity analysis w.r.t. the number of cells

In Figure 6, we illustrate how InfoSEM with different priors and DeepSEM perform on the unseen gene test sets when only a fraction of cells in the hESC dataset are used to train the model. We observe that InfoSEM-BC is only a slightly better than InfoSEM-B. Moreover, InfoSEM-B and InfoSEM-BC trained only on 20% cells can achieve a similar performance as DeepSEM trained on all cells.

| | hESC | | hHEP | | mDC | | mESC | |
|---|---|---|---|---|---|---|---|---|
| | AUPRC | Hit@1% | AUPRC | Hit@1% | AUPRC | Hit@1% | AUPRC | Hit@1% |
| One-hot LR | 0.025 (0.002) | 0.029 (0.008) | 0.022 (0.002) | 0.006 (0.006) | 0.022 (0.002) | 0.012 (0.004) | 0.014 (0.001) | 0.008 (0.004) |
| MatComp | 0.024 (0.001) | 0.017 (0.004) | 0.021 (0.001) | 0.017 (0.003) | 0.023 (0.001) | 0.031 (0.005) | 0.014 (0.000) | 0.014 (0.003) |
| scGREAT | 0.025 (0.003) | 0.031 (0.011) | **0.029 (0.003)** | 0.027 (0.010) | 0.028 (0.002) | 0.024 (0.006) | **0.024 (0.001)** | 0.037 (0.006) |
| GENELink | 0.023 (0.001) | 0.006 (0.006) | 0.020 (0.002) | 0.014 (0.007) | 0.026 (0.002) | 0.028 (0.013) | 0.015 (0.001) | 0.012 (0.003) |
| SCENIC | 0.020 (0.001) | 0.030 (0.007) | 0.025 (0.002) | 0.044 (0.008) | 0.021 (0.001) | 0.016 (0.003) | 0.022 (0.003) | **0.042 (0.009)** |
| GRNBoost2 | 0.022 (0.001) | 0.018 (0.006) | 0.022 (0.001) | 0.027 (0.009) | 0.024 (0.001) | 0.020 (0.004) | 0.022 (0.003) | 0.041 (0.009) |
| DeepSEM | 0.028 (0.002) | 0.032 (0.005) | 0.026 (0.003) | 0.028 (0.010) | 0.026 (0.001) | 0.028 (0.004) | 0.022 (0.002) | 0.028 (0.007) |
| InfoSEM-B (Ours) | **0.036 (0.004)** | **0.038 (0.007)** | **0.029 (0.003)** | **0.049 (0.009)** | **0.050 (0.004)** | **0.077 (0.014)** | **0.023 (0.003)** | 0.032 (0.008) |
| InfoSEM-BC (Ours) | **0.038 (0.004)** | **0.042 (0.006)** | **0.030 (0.003)** | **0.054 (0.007)** | **0.051 (0.004)** | **0.082 (0.014)** | **0.023 (0.002)** | **0.045 (0.007)** |
| Random | 0.024 | 0.024 | 0.021 | 0.021 | 0.024 | 0.022 | 0.014 | 0.014 |

*Table 6.* The GRN inference performance for each method evaluated on *interactions between unseen genes* with **non-cell-type specific ChIP-seq** targets. The top-2 best models are **bold**. We observe that InfoSEM-BC achieves top-2 performance on all dataset and InfoSEM-B achieves top-2 on three datasets.

| | hESC | hHEP | mDC | mESC |
|---|---|---|---|---|
| DeepSEM | 0.10 (0.04) | 0.11 (0.04) | 0.60 (0.03) | 0.03 (0.02) |
| InfoSEM-B (Ours) | 0.19 (0.09) | 0.34 (0.14) | 0.64 (0.10) | 0.19 (0.12) |
| InfoSEM-BC (Ours) | 0.22 (0.09) | 0.38 (0.14) | 0.64 (0.10) | 0.21 (0.11) |

*Table 7.* Averaged recall of each method with non-cell-type specific target GRNs using a threshold 0.5.

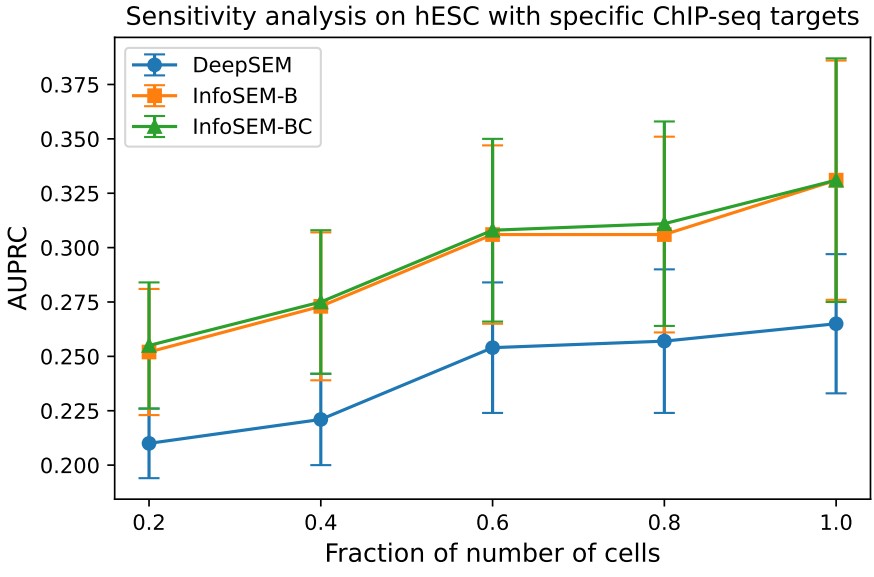

*Figure 6.* AUPRC of models on the *unseen genes* test set with different numbers of cells in the training data.

