# OpenReview forum: "InfoSEM: A Deep Generative Model with Informative Priors for Gene Regulatory Network Inference"
_ICML.cc/2025/Conference — ICML 2025 poster_

### Official Review · Reviewer_TKSk · 2025-03-10

**Overall Recommendation:** 3

**Summary:**

This paper emphasizes the reason why unsupervised gene regulatory network inference lags in supervised ones is that it did not well utilize the prior knowledge. They followed the framework proposed by DeepSEM and proposed infoSEM to involve prior knowledge provided by BioBERT and logit probability. In addition, the paper argued that the previous benchmarks suffer from robustness and imbalanced classes. They improved it by making sure that genes in test sets are not in training sets. The effectiveness of InfoSEM is verified on real-world datasets with Ground-Truth provided by ENCODE and  ChIP-Atlas.

**Claims And Evidence:**

This paper relies on an important assumption that genes, which are similar in function, prefer to have similar causal relationships. I think this claim needs evidence to support it. I do not know if it is reasonable in biology.

**Essential References Not Discussed:**

No

**Experimental Designs Or Analyses:**

Yes. I am curious about how to align the splitting of test sets and the ground truth in evaluation, as if you randomly split the training and test sets, some genes may not be reported by Chip-seq.

Usually the sc-RNA seq datasets have around 5000 genes, why the dataset you choose all is around 1000 genes.

**Methods And Evaluation Criteria:**

I agree with that Chip-seq is used as ground truth to evaluate the performance. But ENCODE and ChIP-Atlas usually have limited records, how do you evaluate the output gene regulatory network?

1. How can we guarantee that the prior knowledge from BioBERT is reliable?
2. For gene regulatory network inference, if you train a model for different cells, how do you guarantee that the output network is suitable for all cells, as the expressed genes are different among different cells?
3. If you consider the data generation process following e.q. (3), how to identify the direction of genes if there is a latent variable affecting them?

**Other Comments Or Suggestions:**

No

**Other Strengths And Weaknesses:**

How to guarantee the asymmetry of gene regulatory network, especially, considering the confounding effect provided by latent variables?

**Questions For Authors:**

No

**Relation To Broader Scientific Literature:**

Providing a way to involve prior knowledge for unsupervised learning.

**Theoretical Claims:**

There are no theoretical claims

---

> ### Author Rebuttal · Authors · 2025-03-31
>
> We appreciate the reviewer’s insightful comments. Below, we address their questions.
>
> **1. Do Functionally Similar Genes Share Causal Relationships?**: The assumption that functionally similar genes share causal relationships is well supported by biological literature. Evolutionary principles suggest that genes used for related functions often maintain similar regulatory architectures [1,2]. Mendelian randomization studies show that genes affecting related traits frequently share causal variants in lipid metabolism and coronary heart disease [3,4]. Pleiotropy analyses further demonstrate that cell-type-specific genes influence multiple neuropsychiatric traits via overlapping pathways, reinforcing the notion of shared causal roles among functionally related genes [5]. We will include this discussion and relevant citations in the camera-ready version.
>
> **2. Evaluation of the Output GRN**: We use the established BEELINE GRN evaluation dataset [7], which only evaluates interactions with recorded ground-truth labels, consistent with previous works [8,9].
>
> **3. Reliability of BioBERT-Derived Prior**: BioBERT has been shown to enhance scRNA-seq information for GRN inference in prior work [10]. In our InfoSEM framework, we treat BioBERT embeddings as an additional prior, with their influence adjustable through a cross-validated hyperparameter that allows the model to reduce their importance if deemed unreliable.  Still, to further validate the reliability of BioBERT embeddings in response to the reviewer's question, we performed Gene Ontology enrichment analysis on gene clusters derived from hierarchical clustering of the similarity matrix of BioBERT embeddings, controlling for false discovery rate (FDR). Analysis of 908 genes revealed several significant functional enrichments post-FDR correction (FDR < 0.05), notably in lipid transport (FDR=6.80e-12), DNA replication initiation (FDR=6.95e-07), and RNA polymerase II transcription (FDR=3.49e-07). The enrichment patterns displayed clear biological specificity; for instance, one cluster included APOA1, APOA2, APOA4, and APOB, enriched for cholesterol efflux (FDR=4.12e-14), while another cluster encompassed multiple MED genes related to transcription regulation. These robust enrichments strongly suggest that BioBERT embeddings capture meaningful biological relationships between genes.
>
> **4. Training Single Model for Different Cell Types**:  This appears to be a misunderstanding. We do not train a single model for multiple cell types. Like DeepSEM [8] and other baselines, we train separate models for each cell type to capture cell-type-specific regulatory interactions instead of assuming a universal gene regulatory network.
>
> **5. Handling Latent Variables & Network Asymmetry**: The impact of latent variables on inferred direction of GRNs is an important research area, as shown in studies like [6]. We will include a discussion of this in our paper and cite [6]/other relevant works. However, our study relies on established datasets with experimentally validated interactions from prior works [7], which we believe minimizes the impact of latent confounders. Addressing confounders comprehensively will be a non-trivial task that warrants a separate paper, beyond the scope of our work.
>
> **6. Alignment of Test Set Split and ChIP-seq Ground Truth**: Like question 2's response, we adhere to established literature by considering only known interactions from ChIP-seq databases [7,8,9] and splitting them into training and test sets.
>
> **7. Choice of 1000 Genes Instead of 5000**: We follow BEELINE's established preprocessing method [7], selecting the top 1000 highly expressed genes to enhance the quality of inferred networks and ensure comparability with prior methods [8,9].
>
> In light of additional experiments & clarifications provided in response to reviewer's comments, we kindly request consideration on raising the score. We believe these clarifications address all the comments!
>
> #### Reference
> [1] Evaluating the potential role of pleiotropy in Mendelian randomization studies, 2018
> [2] Inferring causality and functional significance of human coding DNA variants, 2012
> [3] Diagnostics for Pleiotropy in Mendelian Randomization Studies: Global and Individual Tests for Direct Effects, 2018
> [4] Pleiotropy-robust Mendelian randomization, 2018
> [5] Selecting causal genes from genome-wide association studies via functionally-coherent subnetworks, 2016
> [6] Gene Regulatory Network Inference in the Presence of Selection Bias and Latent Confounders, 2025
> [7] Benchmarking algorithms for gene regulatory network inference from single-cell transcriptomic data, 2020 \
> [8] Modeling gene regulatory networks using neural network architectures, 2021 \
> [9] Graph attention network for link prediction of gene regulations from single-cell RNA-sequencing data, 2022 \
> [10] scGREAT: Transformer-based deep-language model for gene regulatory network inference from single-cell transcriptomics, 2024

---

> > ### Comment · Reviewer_TKSk · 2025-04-03
> >
> > Thanks for the author's response!
> >
> > **Re:1:** If this assumption is common or can be a principle to follow if only supported by some cases, I think it is not enough. Another concern is whether the similarity in the embedding of BioBert can accurately represent functional consistency instead of only similarities.
> >
> > **Re4 and 6** Differential gene expression is a well-established concept in biology. Based on this phenomenon, it is commonly assumed that gene regulatory networks (GRNs) are context-specific—meaning that different cell lines may have distinct underlying regulatory mechanisms. You mentioned using ChIP-seq data as a reference; however, I believe it is challenging to ensure consistency due to the limited availability and coverage of ChIP-seq records. So I still have concerns about the ground truth. Could you introduce more about which ChIP-seq data you use, and how consistence between the ChIP-seq data and ScRNA-seq data?

---

> > > ### Author Response · Authors · 2025-04-03
> > >
> > > Thank you! We provide additional clarifications below.
> > >
> > > ## Functionally Similar Genes Share Causal Relationships
> > >
> > > We reiterate that functionally similar genes sharing causal relationships is not a hypothesis, but rather a well-established biological fact supported by extensive literature with insights from statistical genetics and evolutionary principles. In our rebuttal, we cited five papers [1-5] supporting this and here we add two more works further reinforcing this fact [11-12]. We will include them in related works section in paper.
> > >
> > > We believe the reviewer's claim that this fact is "only supported by some cases" overlooks the substantial evidence we have cited, which consistently demonstrates these relationships across numerous independent studies. Given the scientific evidence, we see no reason to dispute the validity of this fact and believe our expanded references clarify it sufficiently.
> > >
> > > ## Similarity in BioBERT embedding accurately represents functional consistency
> > >
> > > As mentioned in our rebuttal, firstly **our method utilizes BioBERT gene embeddings as a flexible prior while learning the appropriate weights to assign to this prior and hence can automatically down-weight their contribution, if deemed unreliable.**
> > >
> > > Secondly, in our rebuttal we already provided further **evidence of BioBERT embeddings' effectiveness in representing functional consistency via gene-set enrichment analysis**, a well-established technique in biology cited over 50,000 times [13]. Conducting Gene Ontology enrichment on gene clusters from BioBERT embeddings revealed strong functional enrichments in processes such as lipid transport, DNA replication initiation, and RNA polymerase II transcription, demonstrating clear biological specificity in clusters derived from BioBERT gene embeddings and consistent functional relationships among similar genes.
> > >
> > > Importantly, **our main contribution lies not in the use of BioBERT embeddings as prior, but rather in InfoSEM, a variational Bayes framework for GRN inference that can incorporate any informative gene embedding to enhance performance. We showed that other effective priors, such as GenePT embeddings (in main text) and Gene Ontology knowledge graph embeddings (in our rebuttal to Reviewer bvmZ), also improve GRN inference with our method.** Therefore, if other gene embeddings are identified to be more reliable, they can be seamlessly integrated in place of BioBERT embeddings, as evidenced by our experiments.
> > >
> > > ## GRNs are context-specific—different cell lines may have distinct underlying regulatory mechanisms
> > >
> > > We fully agree with reviewer on the importance of cell-type specificity and **this is precisely why we train separate models for different cell types**, a practice also adopted by others in the field. The cell-type specificity of GRN is demonstrated through our cell-type-specific target experiments in Table 2 in paper already. We are not aware of any instance in the paper or rebuttal that contradicts this point.
> > >
> > > ## More about ChIP-seq data/consistency between ChIP-seq and scRNA-seq
> > >
> > > We use scRNA-seq and ChIP-seq data from BEELINE project [7] (Pratapa et al., Nature Methods, 2020), which collected and preprocessed scRNA-seq data from five different cell lines (two human, three mouse) with corresponding ChIP-seq data from **the same or similar cell types for each scRNA-seq dataset**, sourced from ENCODE, ChIP-Atlas, and ESCAPE databases. For detailed information, refer to "Datasets" section of BEELINE paper (Nature Methods version). The datasets have been widely recognized for cell-type specific GRN inference, as evidenced by their use in literature [8-9, 14] and highlighted in three more works by Reviewer TUBJ [15-17]. Notably, BEELINE paper and its associated datasets have been cited over 680 times, reflecting their quality and reliability.
> > >
> > > While acquiring more comprehensive ChIP-seq datasets for multiple cell lines is valuable, it is beyond the scope of our ML-focused paper, which utilizes existing public datasets to propose new ML models.
> > >
> > > We believe this addresses all questions and we request the reviewer to consider raising their score in light of these clarifications!
> > >
> > > ## References:
> > > References [1-10] mentioned in original rebuttal \
> > > [11] Shared associations identify causal relationships between gene expression & immune cell phenotypes, 2021 \
> > > [12] Shared genetic architecture and causal relationship between frailty & schizophrenia, 2025 \
> > > [13] Gene set enrichment analysis: a knowledge-based approach for interpreting genome-wide expression profiles, 2005 \
> > > [14] DeepDRIM: a deep neural network to reconstruct cell-type-specific GRN using scRNA-Seq data, 2021 \
> > > [15] Boosting single-cell GRN reconstruction via bulk-cell transcriptomic data, 2022 \
> > > [16] GRN Inference from Pre-trained Single-Cell Transcriptomics Transformer with Joint Graph Learning, 2024 \
> > > [17] STGRNS: an interpretable transformer-based method for inferring GRNs from single-cell transcriptomic data, 2023

---

### Official Review · Reviewer_rrUh · 2025-03-14

**Overall Recommendation:** 4

**Summary:**

This paper introduces a method that incorporates gene embeddings from pretrained language models or known gene-gene relationships into the existing DeepSEM framework, resulting in two models: InfoSEM-B and InfoSEM-BC. The authors provide a detailed discussion on how to integrate the interaction matrix $Y$ and adjacency matrix $A$ when introducing gene relationships. Additionally, they propose a new benchmark approach for evaluating gene regulatory network model performance, where not only are interactions absent from the training set, but transcription factors are also excluded from the training set.

**Claims And Evidence:**

Yes, the authors support their claims through comparisons with baselines and through precise ablation experiments and relevant analyses.

**Essential References Not Discussed:**

See the Familiarity With Literature and Questions For Authors sections.

**Experimental Designs Or Analyses:**

I have reviewed the performance comparison tables with baselines and the ablation experiment tables. The analysis results demonstrate the effectiveness of the author's proposed method and the effectiveness of using gene embeddings.

**Methods And Evaluation Criteria:**

Yes, AUPRC and Hit@1% appear to be reasonable metrics for evaluating the accuracy of gene regulatory network predictions.

**Other Comments Or Suggestions:**

See Questions For Authors.

**Other Strengths And Weaknesses:**

This paper appears to be an incremental improvement on DeepSEM. Nonetheless, I personally believe that the work of integrating external gene knowledge and gene relationships is quite important. The authors' experimental conclusions demonstrate the effectiveness of these biological priors in improving gene regulatory network prediction performance, which I think may inspire more work in this direction.

**Questions For Authors:**

1. In Section 2.2, do the mentioned external priors specifically refer to the gene embeddings and gene relationships mentioned later? Could you provide a brief explanation here?
2. Would the authors have time to add a comparison with the SCENIC method?
3. How does the authors' proposed model generalize to the $D_\text{unseen}$ set? Is it through gene embeddings and gene interactions?
4. Why did you prioritize trying gene embeddings from pretrained language models? How effective would gene embeddings from RNA foundation models (such as scGPT, scFoundation, etc.) be?
5. Did the authors consider the impact of data leakage when integrating external priors? Were any validation measures taken?

**Relation To Broader Scientific Literature:**

The authors' proposed method is a direct extension of the DeepSEM[1] method published in 2021.

[1] Modeling gene regulatory networks using neural network architectures

**Theoretical Claims:**

I have examined the author's explanations regarding the variational inference proofs in the main text and did not find any issues.

---

> ### Author Rebuttal · Authors · 2025-03-31
>
> We thank the reviewer for their thoughtful and constructive comments. We appreciate the recognition of our new benchmark and the value of integrating external priors for GRN inference. We address the reviewer’s questions below and will include them in the camera-ready version.
>
> **1. External Priors in Section 2.2**: Yes, the external priors refer to gene embeddings and gene relationships. These priors can be derived from various sources, including textual embeddings (e.g., BioBERT), knowledge graphs (e.g., Gene Ontology), or known gene interactions, all of which original DeepSEM cannot use directly.
>
> **2. Comparison with SCENIC**: We have now included a comparison with SCENIC, which demonstrates similar performance as GRNBoost2. Our InfoSEM-B and InfoSEM-BC outperform SCENIC across all cell lines for AUPRC and across all but hHEP for Hit@1% metric for the cell-type specific datasets.
>
> | Method| hESC AUPRC | hESC Hit@1% | hHEP AUPRC | hHEP Hit@1% | mDC AUPRC | mDC Hit@1% | mESC AUPRC | mESC Hit@1% |
> |-------|----------- |-------------|------------|-------------|-----------|------------|------------|-------------|
> | SCENIC| .210(.020)| .200(.037)| .465(.020)| **.568(.047)**| .227(.014)| .219(.062)| .346(.024)| .393(.045)|
> | InfoSEM-B (Ours)| **.331(.055)**| .547(.091)| **.498(.020)**| .533(.048)| .298(.028)| .472(.076)| .388(.023)| .522(.047)|
> | InfoSEM-BC (Ours)| **.331(.056)**| **.585(.094)**| **.499(.020)**|.550(.053)| **.322(.032)**| **.498(.069)**| **.408(.020)**| **.575(.045)**|
>
> **3. Reason for InfoSEM generalization to D_{unseen}**:  InfoSEM's main training objective is the gene expression reconstruction rather than the interaction prediction used in the supervised learning framework. This encourages the model to learn the gene-gene relationships from the gene expression data, instead of utilizing the gene-specific bias from the labels, hence contributing to its ability to generalize to the unseen gene set, D_{unseen}. Additionally, using gene embeddings as priors aids our model's generalization to the unseen gene set by leveraging those priors to learn gene relationships.
>
> **4. Use of Gene Embeddings from Single-cell Foundation Models**: We initially focused on gene embeddings from pretrained language models like BioBERT due to their effectiveness in providing semantic context for various models and tasks (see [1,2]). Furthermore, BioBERT is easier to use; it requires only the gene names already available, unlike single-cell models that necessitate specific preprocessing of scRNA-Seq data.
>
> Following the reviewer's suggestion, we experimented with embeddings from scBERT [3], a foundation model designed for single-cell RNA data, but did not observe significant improvements in performance over InfoSEM-B working with a one-hot baseline instead. We hypothesize that this may result from scBERT's binning of scRNA-Seq data, which could lead to a loss of important information. Additionally, scBERT's embeddings, derived from binned scRNA-Seq data, are similar to those used in InfoSEM already and may not provide complementary insights for the adjacency matrix compared to embeddings like Gene Ontology or BioBERT, which are derived independent of scRNA-Seq data. We believe, though, that exploring other scRNA-Seq foundation model derived gene embeddings presents an interesting avenue for future exploration.
>
> In response to a similar query from reviewer bvmZ regarding the use of external priors from knowledge graphs, we conducted additional experiments using Gene Ontology (GO) embeddings as priors. The AUPRC results, shown below on the unseen-genes benchmark, demonstrate that this further enhances our InfoSEM results on human cell lines, reinforcing the importance of incorporating external priors from independent sources.
>
> | Method  | hESC AUPRC | hHEP AUPRC|
> |---------|------|------|
> | DeepSEM| .265(.032)| .435(.019)|
> | Our InfoSEM-B (onehot) | .270(.038)| .447(.027)|
> | Our InfoSEM-B (BioBERT) | .332(.055)| .498(.020)|
> | Our InfoSEM-B (GO) | **.345(.057)**|**.524(.034)**|
> | Our InfoSEM-B (scBERT) | .272(.029)| .442(.027)|
>
> **5. Data Leakage and Validation Measures**: Thank you for the important remark! Unlike seen-gene benchmarks, we ensured there was no data leakage between training and test sets in our proposed unseen benchmarking setup, as the genes between training and testing are different.
>
> We hope our clarifications and additional experiments answer all your valuable comments. In light of the additional experiments and results presented, we kindly ask if the reviewer would consider raising their score. Thank you!
>
> #### References
> [1] scGREAT: Transformer-based deep-language model for gene regulatory network inference from single-cell transcriptomics, 2024 \
> [2] GenePT: A Simple But Effective Foundation Model for Genes and Cells Built From ChatGPT, 2024 \
> [3] scBERT as a large-scale pretrained deep language model for cell type annotation of single-cell RNA-seq data, 2022

---

> > ### Comment · Reviewer_rrUh · 2025-04-04
> >
> > Thank you for your detailed and thoughtful response. Your clarifications and additional experiments have addressed my concerns thoroughly. Based on your response and the updated results, I have raised my score accordingly.

---

### Official Review · Reviewer_TUBJ · 2025-03-14

**Overall Recommendation:** 3

**Summary:**

Summary

This study proposes infoSEM, a deep generative model with informative priors for gene regulatory network (GRN) inference. By integrating text-based gene embeddings as biological priors, it addresses the critical challenge of GRN reconstruction without ground-truth interaction labels. The authors also introduce a novel benchmarking framework to evaluate predictions for unseen gene relationships.

Main findings:

1. Leveraging textual gene embeddings as informative priors enhances GRN inference when interaction labels are unavailable.

2. Further integrate available interaction labels as additional priors to reduce bias and improve performance.

Main Results:

1. Evaluated the performance of existing supervised and unsupervised methods under established benchmarks, analyzing why simple methods like one-hot logistic regression (LR) perform remarkably well.

2. Supervised models (e.g., scGREAT and GENELink) showed significant performance degradation in predicting interactions between unseen genes, while unsupervised methods (especially DeepSEM) outperformed supervised methods across three datasets.

3. The InfoSEM-B model improves GRN inference by 25% and 52% on average over DeepSEM when gene interactions are unknown. The InfoSEM-BC variant, which incorporates known interaction information, achieves the best AUPRC performance across all datasets.

4. The study also demonstrates that BioBERT embeddings enhance GRN inference.


Main algorithmic ideas:

Built upon the DeepSEM framework, integrating BioBERT embeddings and gene interaction embeddings as prior knowledge.

**Claims And Evidence:**

1. Under existing benchmarks, the performance of current supervised and unsupervised methods is significantly inferior to the one-hot logistic regression (LR) approach. The minimal differences in AUPRC also were observed on the new benchmark dataset. A question arises: Would replacing one-hot encoding with BioBERT embeddings in LR yield improved performance on the new benchmark?


2. InfoSEM-B and InfoSEM-BC were only evaluated on the new benchmark, with no assessment conducted on original benchmarks.

**Essential References Not Discussed:**

The paper adequately cites and discusses foundational works relevant to GRN inference, including:

1. Supervised approaches(scGREAT, GENELink)
2. Unsupervised variational approaches (DeepSEM, GRNBoost2)

While the core references are sufficient, expanding the discussion to include:

1. Gene Regulatory Network Inference from Pre-trained Single-Cell Transcriptomics Transformer with Joint Graph Learning. ICML 2024 AI for Science workshop

2. Boosting single-cell gene regulatory network reconstruction via bulk-cell transcriptomic data. Briefings in Bioinformatics, 23(5):bbac389, 2022.

3. STGRNS: an interpretable transformer-based method for inferring gene regulatory networks from single-cell transcriptomic data. Bioinformatics 2023

**Experimental Designs Or Analyses:**

The experimental design (unseen gene) is reasonable.



Experimental analysis:

1. InfoSEM-B and InfoSEM-BC only evaluate the results on the new benchmark, not the results on the original benchmark.

2. Ablation studies should be supplemented, such as BioBert's dembedding + LR, ...

**Methods And Evaluation Criteria:**

1. The proposed method demonstrates limited technical improvements compared to DeepSEM, primarily consisting of additional embedding incorporation.

2. The newly proposed benchmark represents a significant advancement for GRN inference research, particularly in addressing the critical challenge of generalizability to unseen gene relationships.

**Other Comments Or Suggestions:**

None

**Other Strengths And Weaknesses:**

Strengths:

1. Conducts systematic evaluation of supervised vs unsupervised methods across legacy and novel benchmarks

2. Empirically demonstrates BioBERT's effectiveness as biological priors for GRN inference.


Weaknesses:

1. The proposed method shows limited architectural innovation over DeepSEM (primarily embedding additions)

2. InfoSEM variants lack ablation studies

**Questions For Authors:**

1. The proposed method demonstrates limited technical improvements compared to DeepSEM, primarily consisting of additional embedding incorporation.


2. Under existing benchmarks, the performance of current supervised and unsupervised methods is significantly inferior to the one-hot logistic regression (LR) approach. The minimal differences in AUPRC also were observed on the new benchmark dataset. A question arises: Would replacing one-hot encoding with BioBERT embeddings in LR yield improved performance on the new benchmark?

3. InfoSEM-B and InfoSEM-BC were only evaluated on the new benchmark, with no assessment conducted on original benchmarks.

4. Ablation studies should be supplemented, such as different loss (VAE -> AE), and more other variants.

5. While the core references are sufficient, expanding the discussion to include:
[1] Gene Regulatory Network Inference from Pre-trained Single-Cell Transcriptomics Transformer with Joint Graph Learning. ICML 2024 AI for Science workshop
[2] Boosting single-cell gene regulatory network reconstruction via bulk-cell transcriptomic data. Briefings in Bioinformatics, 23(5):bbac389, 2022.
[3] STGRNS: an interpretable transformer-based method for inferring gene regulatory networks from single-cell transcriptomic data. Bioinformatics 2023

**Relation To Broader Scientific Literature:**

This work built upon the DeepSEM framework, integrating BioBERT embeddings and gene interaction embeddings as prior knowledge.

**Theoretical Claims:**

1. The work provides detailed theoretical derivations and mathematical formulations, enabling readers to easily comprehend the mathematical principles and logical foundations underlying the model.

2. Existing methodologies are thoroughly described and systematically categorized.

---

> ### Author Rebuttal · Authors · 2025-03-31
>
> We appreciate the reviewer’s feedback and acknowledgment of our real-world benchmark as a key advancement in GRN inference, along with their praise for our theoretical clarity. Below, we address their questions and provide additional results, which we will include in the camera-ready version.
>
> **1. Ablation studies (BioBERT embeddings+LR, AE loss)**:
> Thank you for the suggestion! We have now tested BioBERT embeddings in LR on unseen genes benchmark. Results below show that BioBERT+LR performs similarly or slightly better than the 1-hot LR baseline in paper. This is expected, as in the unseen-gene setting, both LR baselines (with 1-hot or BioBERT) can't leverage gene-specific biases for GRN inference. The slight improvement with BioBERT embeddings highlights the informativeness of the embeddings over 1-hot encoding. However, both these baselines perform considerably worse compared to our InfoSEM-B/InfoSEM-BC models, underscoring the benefits of our proposed framework as a whole beyond simply incorporating BioBERT as prior.
>
> We explored the AE loss function (DeepSEM-AE) suggested by the reviewer, but it performed significantly worse than DeepSEM and our InfoSEM-B and InfoSEM-BC models due to DeepSEM's assumption of a Gaussian distribution for **Z** [1, 2], which requires regularization in the variational Bayes loss. Our InfoSEM-B and InfoSEM-BC naturally serve as ablation studies on the baseline DeepSEM. InfoSEM-B serves as an ablation study by replacing DeepSEM's Laplace prior over A with our informative prior, while InfoSEM-BC adds **Aˡ** and its informative prior to InfoSEM-B. We are open to any further ablation studies if the reviewer suggests.
>
> |Method|hESC AUPRC|hESC Hit@1%|hHEP AUPRC|hHEP Hit@1%|mDC AUPRC|mDC Hit@1%|mESC AUPRC|mESC Hit@1%|
> |------------|------------|-------------|------------|-------------|-----------|------------|------------|-------------|
> | 1-hot LR |.210(.018)| .205(.041)| .395(.016)| .345(.056)| .247(.019)| .225(.004)| .329(.026)| .397(.036)|
> | BioBERT+LR|.212(.019)| .227(.050)| .423(.023)| .450(.076)| .230(.020)| .151(.050)| .347(.021)| .397(.088)|
> | DeepSEM-AE|.203(.016)| .183(.033)| .400(.018)| .483(.054)| .211(.009)| .143(.022)| .321(.021)| .311(.031)|
> | DeepSEM| .265(.032)| .419(.059)| .435(.019)| .517(.043)| .277(.014)| .292(.095)| .343(.024)| .369(.048)|
> | InfoSEM-B (Ours)|**.331(.055)**| .547(.091)| **.498(.020)**| .533(.048)| .298(.028)| .472(.076)| .388(.023)| .522(.047)|
> | InfoSEM-BC (Ours)|**.331(.056)**| **.585(.094)**| **.499(.020)**| **.550(.053)**| **.322(.032)**| **.498(.069)**| **.408(.020)**| **.575(.045)**|
>
> **2. Evaluation on Original Benchmarks**:
> Thank you for your comment! We have now included InfoSEM on the original seen-gene benchmarks in our response to reviewer bvmZ, with results presented in the accompanying table. Our results show that for traditional seen-gene benchmarks, simple supervised models (One-hot LR & MatComp) perform best by leveraging gene-specific biases without using scRNA-seq data, a fact which our work highlights for the first time and is a key contribution of our work. Even in this context, InfoSEM-BC maintains a competitive balance, avoiding overfitting gene-specific biases while performing better than unsupervised baselines. However, the key takeaway is that for seen-gene benchmarks, simpler models that exploit gene-specific biases are sufficient, and hence, we propose to evaluate advanced machine learning models on unseen-gene benchmarks.
>
> **3. Limited  architectural innovation over DeepSEM**:
> We emphasize that our contribution is not on architectural innovation over DeepSEM, but in extending it within a principled variational Bayes framework to leverage informative priors (e.g., gene embeddings from language models) or available ground truth information without overfitting to gene-specific biases, an aspect not explored so far. Additionally, we show for the first time that existing supervised learning models tend to leverage gene-specific biases in the dataset, and hence, we introduce a new unseen genes benchmark to provide a reliable testbed for evaluating generalizability of GRN inference methods. As the reviewer themselves noted, our benchmark for evaluating generalizability to unseen gene relationships is a crucial advancement for GRN inference, and we are happy that it has been well received.
>
> **4. Including citations to proposed references**: We thank the reviewer for additional references. We have already cited the first and third references in the paper and will include the second one in the camera-ready version too. These citations will enhance the discussion and provide further context to our work.
>
> We hope that our clarifications and additional results address your concerns! We kindly request the reviewer to consider raising their score, taking these into account.
>
> #### References
> [1] DAG Structure Learning with Graph Neural Networks, 2019 \
> [2] Modeling gene regulatory networks using neural network architectures, 2021

---

### Official Review · Reviewer_bvmZ · 2025-03-17

**Overall Recommendation:** 3

**Summary:**

The paper introduces InfoSEM, an unsupervised generative model for Gene Regulatory Network (GRN) inference that leverages textual gene embeddings as informative priors. The model can also integrate ground truth (GT) labels when available, avoiding biases and enhancing performance. The authors propose a new benchmarking framework that evaluates interactions between unseen genes, better aligning with real-world applications like biomarker discovery. The paper also highlights limitations in existing supervised learning benchmarks, showing that supervised models may exploit dataset biases rather than capturing true biological mechanisms.

**Claims And Evidence:**

The authors claim that current GRN inference benchmarks fail to generalize well because they are trained and evaluated on the same set of genes, leading to an overestimation of model performance. This claim is supported by experiments where supervised models show significant performance drops when evaluated on unseen genes.

However, a major limitation is that the proposed method is only evaluated on the new unseen-gene benchmark, and it does not compare performance in the traditional setting where models infer unseen regulatory interactions among known genes (seen-gene sets). Given that many real-world biological applications prioritize discovering novel interactions among well-studied genes, the broader impact of InfoSEM’s approach may be limited.

**Essential References Not Discussed:**

The paper should discuss related work in knowledge graph-based GRN inference, as BioBERT embeddings are not the only way to encode biological prior knowledge.

**Experimental Designs Or Analyses:**

The experiments convincingly demonstrate that supervised models overfit seen genes and fail to generalize to unseen genes.

However, the paper does not compare InfoSEM’s performance for discovering unseen interactions among seen genes, which is a key limitation. In many real-world settings, researchers seek to uncover new regulatory interactions within well-characterized gene networks, rather than predicting interactions for entirely unseen genes.

Computational efficiency is not addressed, which is critical for scaling deep generative models to larger genomic datasets.

**Methods And Evaluation Criteria:**

The evaluation methodology is well-structured, incorporating multiple datasets from the BEELINE suite and relevant baselines (e.g., DeepSEM).

However, the paper only evaluates InfoSEM on the new unseen-gene benchmark and does not compare its performance in the traditional seen-gene setting. This omission raises concerns about whether InfoSEM is broadly applicable to all GRN inference tasks.

The rationale for selecting BioBERT embeddings as informative priors is reasonable, but it would be helpful to compare against other gene representation techniques (e.g., knowledge graph embeddings).

**Other Comments Or Suggestions:**

The authors should evaluate InfoSEM’s performance in the seen-gene setting to determine whether it is broadly applicable.
A runtime comparison of InfoSEM vs. baselines would improve understanding of its computational feasibility.
A more explicit discussion of the real-world implications of predicting interactions among unseen genes vs. seen genes would strengthen the paper.

**Other Strengths And Weaknesses:**

Strengths:
Novel use of BioBERT embeddings as informative priors for GRN inference.
Insightful critique of dataset biases in supervised GRN benchmarks.
Demonstrates strong performance on the new unseen-gene benchmark.

Weakness:
Does not evaluate InfoSEM in the traditional seen-gene setting, limiting its applicability to broader GRN inference tasks.
Computational efficiency is not addressed, which is important for scalability.
Lack of theoretical justification for the adjacency matrix decomposition (Ae and Al).

**Questions For Authors:**

1.	Why is InfoSEM not evaluated in the traditional seen-gene setting? This is a crucial benchmark for real-world GRN inference.
2.	How does InfoSEM compare computationally to other deep generative models for GRN inference?
3.	Would using alternative gene embeddings (e.g., knowledge graph embeddings) affect performance?

**Relation To Broader Scientific Literature:**

The paper is well-positioned in the GRN inference and single-cell genomics literature and aligns with recent efforts to develop interpretable AI models for biology.
However, the focus on unseen-gene inference is not well contextualized within broader biological research, where many studies aim to infer novel regulatory interactions among well-characterized genes.

**Theoretical Claims:**

The variational Bayes framework for InfoSEM appears mathematically sound.
The decomposition of the adjacency matrix (Ae and Al) into effect size and interaction probability is novel, but the paper does not provide strong theoretical justification for why this decomposition improves GRN inference.

---

> ### Author Rebuttal · Authors · 2025-03-31
>
> We appreciate the reviewer’s valuable feedback and acknowledgment of our work’s novelty and rigor. We are happy that our analysis of biases in supervised GRN benchmarks and InfoSEM’s performance on the unseen-gene benchmark were well received. Below, we address the reviewer’s questions and provide additional results, which we will include in the camera-ready version of the paper.
>
> **1.Evaluation in Traditional Seen-Gene Setting**: Thank you for the important point! We now show InfoSEM's performance in this setting below for all datasets (cell-type-specific ChIP-seq targets).
>
> | Method     | hESC AUPRC | hESC Hit@1% | hESC AUPRC | hESC Hit@1% | mDC AUPRC | mDC Hit@1% | mESC AUPRC | mESC Hit@1% |
> |------------|------------|-------------|------------|-------------|-----------|------------|------------|-------------|
> | One-hot LR | .600(.028) | .932(.039) | .835(.005) | 1.(.003) | .176(.006) | .250(.024) | .844(.009) | .991(.004)|
> | MatComp    | .638(.026) | .913(.037) | .840(.006) | 1.(.006) | .351(.013) | .419(.034) | .860(.007) | .974(.008) |
> | DeepSEM    | .216(.010) | .318(.016) | .424(.011) | .509(.028) | .118(.006) | .165(.033) | .340(.013) | .431(.023) |
> | InfoSEM-B  | .374(.027) | .752(.067) | .467(.011) | .538(.020) | .130(.010) | .174(.048) | .401(.012) | .562(.016) |
> | InfoSEM-BC | .553(.028) | .913(.053) | .702(.007) | .865(.023) | .285(.016) | .555(.063) | .672(.010) | .918(.026) |
>
> The results show that for traditional seen-gene benchmarks, simple supervised models (One-hot LR, MatComp) without utilizing scRNA-seq perform best by leveraging gene-specific dataset biases, a fact which our work highlights for the first time and is a key contribution of our work. Even in this context, InfoSEM-BC maintains a competitive balance, avoiding overfitting to the gene-specific biases while performing better across multiple datasets compared to other unsupervised methods (GRNBoost2, DeepSEM). However, we again emphasize that the key takeaway is for seen genes benchmark, simple supervised models (One-hot LR, MatComp) that exploit gene-specific biases from known interactions are sufficient even without scRNA-seq. Hence, we propose to evaluate more advanced machine learning models on unseen-gene benchmarks where simple models entirely fail as shown in paper.
>
> **2.Gene embeddings from knowledge graphs**: Thank you for the suggestion! Our flexible framework can incorporate embeddings from other sources, such as knowledge graphs. To showcase this, we now did additional experiments using Gene Ontology (GO)  knowledge graph embeddings as priors (AUPRC shown below on unseen-genes benchmark), which improve our InfoSEM-B (BioBERT) results even further on human cell lines but lag behind our original InfoSEM-B (BioBERT) on mouse cell lines. This is because GO knowledge graph used is specific to human cells. Unlike BioBERT, which is more generic, knowledge graphs have to be chosen carefully depending on the context.
>
> | Method  | hESC | hHEP| mDC | mESC |
> |---------|------|------|------|------|
> | DeepSEM | .265(.032) | .435(.019)| .277(.014)| .343(.024)|
> | Our InfoSEM-B (one-hot)| .270(.038) | .447(.027)| .224(.012)| .338(.022)|
> | Our InfoSEM-B (BioBERT)| .332(.055) | .498(.020)| **.298(.028)**| **.388(.023)**|
> | Our InfoSEM-B (GO)| **.345(.057)** | **.524(.034)**| .275(.011)| .359(.024)|
>
> **3.Compute Efficiency**: We have now included comparison of run times (in seconds) for different configurations of (num_cell, num_gene) for InfoSEM and other baselines. Our results demonstrate that InfoSEM is faster than popular supervised learning baselines and only slightly slower than DeepSEM, while consistently outperforming them in terms of performance in unseen genes benchmark.
>
> | Model     |(454, 844) | (758, 844) | (758, 1291) |
> |-----------|-----------|------------|-------------|
> | scGREAT| 236.30 | 239.96| 357.52 |
> | GENELink| 177.25 | 172.72 | 258.91|
> | SCENIC|  80.81 | 97.74| 142.51|
> | GRNBoost2 |   9.27 | 10.81| 13.74|
> | DeepSEM   |  91.88 | 112.55| 116.01|
> | InfoSEM-B | 118.00 | 134.11| 206.86|
> | InfoSEM-BC| 125.91 | 164.16| 211.37|
>
> **4.Rationale for **Aᵉ** and **Aˡ** splitting of interaction matrix**:
> By splitting the interaction matrix into two components—**Aᵉ** for the magnitude and **Aˡ** for the logit—we create a setup where both components can incorporate prior biological knowledge independently so that any possible misspecification of one prior will not affect the other. The prior on **Aᵉ**, informed by embeddings, acts as a starting point, but the model can adjust its MAP estimate based on data. Similarly, the prior on **Aˡ** allows for a probabilistic treatment of known interactions, but leaves room for learning from the data where these interactions might not be fully observed.
>
> We hope these clarifications and new results address the reviewer’s concerns and enhance our paper's contributions. We kindly request consideration for a score increase based on this. We are happy to address any further suggestions!

---

### Decision · Program_Chairs · 2025-05-01

**Decision:**

Accept (poster)

**Comment:**

This paper proposes an unsupervised generative model for gene regulatory network (GRN) inference incorporating textual gene embeddings from language models as informative priors, improving performance by integrating known gene interactions. A new benchmarking strategy for testing models on unseen genes is introduced, demonstrating that the method outperforms existing supervised and unsupervised methods on multiple real-world datasets.

All reviewers were recommending acceptance of the paper.